# 📈 mTSBench: Benchmarking Multivariate Time Series Anomaly Detection and Model Selection at Scale

**Xiaona Zhou**                                                   *xiaonaz2@illinois.edu*
*Siebel School of Computing and Data Science*
*University of Illinois Urbana-Champaign*

**Constantin Brif**                                               *cnbrif@sandia.gov*
*Center of Computation and Analysis for National Security*
*Sandia National Laboratories*

**Ismini Lourentzou**                                             *lourent2@illinois.edu*
*School of Information Sciences*
*University of Illinois Urbana-Champaign*

**Reviewed on OpenReview:** *https://openreview.net/forum?id=8LfB8HD1WU*

## Abstract

Anomaly detection in multivariate time series is essential across domains such as healthcare, cybersecurity, and industrial monitoring, yet remains fundamentally challenging due to high-dimensional dependencies, the presence of cross-correlations between time-dependent variables, and the scarcity of labeled anomalies. We introduce **mTSBench**, the largest benchmark to date for multivariate time series anomaly detection and model selection, consisting of 344 labeled time series across 19 datasets from a wide range of application domains. We comprehensively evaluate 24 anomaly detectors, including the only two publicly available large language model-based methods for multivariate time series. Consistent with prior findings, we observe that no single detector dominates across datasets, motivating the need for effective model selection. We benchmark three recent model selection methods and find that even the strongest of them remains far from optimal. Our results highlight the outstanding need for robust, generalizable selection strategies. We open-source the benchmark at `https://plan-lab.github.io/mtsbench` to encourage future research.

## 1 Introduction

Multivariate Time Series Anomaly Detection (MTS-AD) is critical for identifying unexpected patterns in multi-signal temporal data across domains such as healthcare, cybersecurity, industrial monitoring, and finance (Blázquez-García et al., 2021; Garg et al., 2021). The rapid digital transformation of these sectors has led to a surge in high-dimensional time series data, where timely and accurate anomaly detection is essential to prevent system failures, mitigate security threats, and optimize operational efficiency (Basu & Meckesheimer, 2007; Sgueglia et al., 2022; Yang et al., 2023). However, identifying anomalies in multivariate time series remains challenging due to their inherent complexity and heterogeneity, compounded by factors such as non-linear temporal relationships, inter-variable correlations, and the sparsity of anomalous events.

A diverse set of approaches to anomaly detection has been developed, including the use of foundation models (Bian et al., 2024; Zhou et al., 2023), deep learning (Xu et al., 2022; Sakurada & Yairi, 2014; Munir et al., 2018; Xu et al., 2018), classic machine learning (Hariri et al., 2019; Goldstein & Dengel, 2012; Yairi et al., 2001), and statistical models (Rousseeuw & Van Driessen, 1999; Hochenbaum et al., 2017). However, the performance of anomaly detection methods varies widely across datasets, with recent works (Braei & Wagner, 2020; Ho et al., 2025; Zamanzadeh Darban et al., 2024; Schmidl et al., 2022; Paparrizos et al., 2022b)

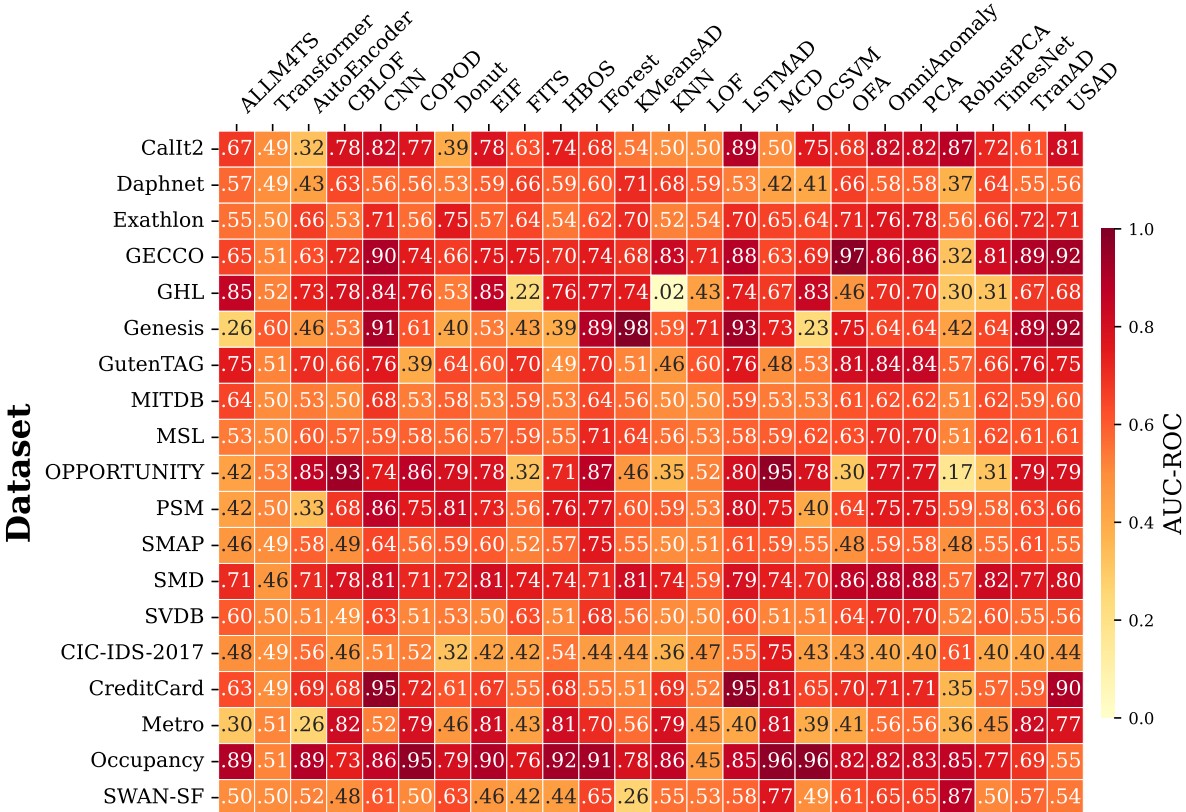

Figure 1: **Average AUC-ROC (↑) Performance of 24 Anomaly Detection Methods (*x*-axis) Evaluated Across 19 mTSBench Datasets (*y*-axis).** The substantial performance variability across datasets highlights the need for robust model selection strategies. **mTSBench** benchmarks the capability of model selection techniques to systematically identify the optimal anomaly detection method among 24 state-of-the-art detectors evaluated on a comprehensive collection of 344 multivariate time series.

emphasizing repeatedly that existing algorithms do not consistently excel in all anomaly detection scenarios. This inconsistency is further compounded by the unsupervised nature of anomaly detection tasks (Mejri et al., 2024; Belay et al., 2023), where ground truth labels are often unavailable, making model selection an open challenge. These challenges in MTS-AD highlight the need for adaptive model selection strategies that can identify the optimal detector for a given multivariate time series dataset.

Existing model selection methods can be grouped into meta-learning, unsupervised, internal evaluation-based, and classifier-based, with many of them leveraging historical performance metrics, meta-features, or internal performance measures to identify the best models for new datasets (Zhao et al., 2021; 2022; Navarro et al., 2023; Zhang et al., 2022b; Goswami et al., 2021). However, their evaluation is often conducted on disparate datasets and tasks, leading to inconsistent and incomparable results across studies. This lack of standardization not only hampers progress in developing effective model selection techniques but also obscures the real-world applicability of proposed methods. Consequently, there is a need for a unified benchmark that systematically evaluates model selection approaches under consistent settings and can facilitate the development of robust selection methods, specifically tailored for the challenging task of MTS-AD.

To address these challenges, we introduce **mTSBench**, the largest and most diverse benchmark for MTS-AD and model selection to date. mTSBench consists of 344 multivariate time series from 19 publicly available datasets, covering 12 application domains. These datasets include both point-based and range-based anomalies, reflecting real-world temporal dependencies and cross-signal interactions. Moreover, the mTSBench evaluation suite spans 24 anomaly detection methods based on various approaches, including reconstruction, prediction, statistics, and large language model (LLM). Our empirical analysis highlights substantial variability in

Table 1: **Comparison Between mTSBench and Existing Anomaly Detection Benchmarks that Contain Multivariate Time Series.** Algorithm categories comprise foundation models (FM), deep learning (DL), classic machine learning (ML) and Other (*e.g.*, statistical, data mining, *etc.*).

| | MTS Data | | Model Selection | | Anomaly Detection | | | | |
|---|---|---|---|---|---|---|---|---|---|
| | # Datasets | # TS | # Selectors | # Metrics | # FM | # DL | # ML | # Other | # Metrics |
| TODS (Lai et al., 2021) | 5 | 25 | ✗ | ✗ | 0 | 2 | 5 | 2 | 3 |
| TIMESEAD (Wagner et al., 2023) | 2 | 21 | ✗ | ✗ | 0 | 26 | 2 | 0 | 3 |
| EEAD (Zhang et al., 2023) | 8 | 8 | ✗ | ✗ | 0 | 6 | 0 | 4 | 4 |
| TIMEEVAL (Schmidl et al., 2022) | 14 | 238 | ✗ | ✗ | 0 | 15 | 7 | 11 | 3 |
| TSB-AD (Liu & Paparrizos, 2024) | 17 | 200 | ✗ | ✗ | 1 | 10 | 7 | 5 | 10 |
| **mTSBench (Ours)** | **19** | **344** | **3** | **3** | **2** | **10** | **7** | **5** | **13** |

anomaly detection performance across datasets. For example, Figure 1 showcases the mean AUC-ROC scores of the 24 detectors across the 19 datasets, illustrating how certain detectors achieve near-perfect accuracy on particular datasets while modest results on others. This inconsistency in detection performance emphasizes the critical importance of model selection strategies that account for dataset characteristics and temporal dependencies. To address this gap, unlike existing benchmarks summarized in Table 1, mTSBench uniquely integrates model selection methods and corresponding evaluation metrics, offering insights into their robustness and adaptability — a crucial aspect of MTS-AD that remains underexplored.

The contributions of our work can be summarized as follows:

**(1)** We introduce **mTSBench**, the largest and most comprehensive MTS-AD and model selection benchmark to date, featuring 344 labeled multivariate time series from 19 datasets across 12 application domains. mTSBench systematically evaluates 24 anomaly detection methods, including the only LLM-based methods for MTS-AD, reflecting their performance in multivariate settings with real-world temporal dependencies and cross-signal interactions.

**(2)** Our empirical analysis reveals that, among the evaluated methods, no single anomaly detection method performs consistently well across datasets in mTSBench, underscoring the need for adaptive selection strategies. To this end, mTSBench is the first to integrate unsupervised model selection methods and benchmark their effectiveness across diverse time series contexts and under consistent settings.

**(3)** To drive reproducible comparisons, mTSBench introduces a unified evaluation suite with point-based and ranking-based metrics for anomaly detection and model selection. Using this standardized setup, we observe substantial gaps between the evaluated unsupervised model selection methods and both optimal and trivial baselines. These results highlight limitations of current unsupervised selection strategies and underscore the need for more adaptive model selection mechanisms.

## 2 Related Work

**Time Series Anomaly Detection.**

Time series data, represented as ordered sequences of real-valued observations, can be categorized into univariate (*e.g.*, single sensor readings) and multivariate (*e.g.*, multiple sensors capturing joint phenomena). Time series anomaly detection is inherently challenging due to the diverse manifestations of anomalies, typically categorized into point anomalies, representing isolated deviations; contextual anomalies, which are abnormal only within a specific temporal context; and collective anomalies, emerging from atypical patterns spanning multiple time steps (Boniol et al., 2024; Blázquez-García et al., 2021; Shaukat et al., 2021; Chandola et al., 2009). Existing surveys have extensively evaluated anomaly detection algorithms across a range of paradigms, including unsupervised (Mejri et al., 2024), explainable (Li et al., 2023), model-based (Correia et al., 2024), and transformer-based approaches (Wen et al., 2023). Dedicated reviews have also covered univariate (Braei & Wagner, 2020; Paparrizos et al., 2022b; Freeman et al., 2021), graph-based (Jin et al., 2024a; Ho et al., 2025), and deep learning-based methods (Zamanzadeh Darban et al., 2024; Yan et al., 2024; Chalapathy & Chawla, 2019), with recent efforts turning to foundation models (Ye et al., 2024; Su et al., 2024; Jin et al., 2024b). Recent work (Schmidl et al., 2022) performed the most comprehensive evaluation to date, analyzing 71 anomaly detection methods across both univariate and multivariate time series. All these studies consistently affirm that no single

anomaly detection method excels universally across domains or anomaly types, motivating the need for dynamic model selection strategies that can adaptively identify the best-performing anomaly detector for a given time series instance. However, while these works provide valuable insights, they largely focus on univariate settings, leaving critical gaps in understanding how detection methods generalize to complex multivariate dependencies.

**Model Selection for Time Series.** Recent advances have led to the development of diverse model selection frameworks that leverage intrinsic dataset properties, meta-learning signals, or evaluation heuristics to find the most suitable anomaly detector for a given time series (Sylligardos et al., 2023; Valenzuela et al., 2024; Trirat et al., 2024; Eldele et al., 2021). Model selection strategies can be broadly categorized into four main approaches: (1) Classifier-based methods, which train classifiers to rank detectors based on extracted meta-features (Ying et al., 2020; Chatterjee et al., 2022); (2) Meta-learning approaches, which predict model performance on unseen datasets by generalizing from historical observations (Singh & Vanschoren, 2022; Yu et al., 2022; Zhao et al., 2021; Navarro et al., 2023; Zhang et al., 2022b); (3) Reinforcement learning-based methods, which optimize detector choice by iteratively adjusting based on feedback signals (Wang et al., 2022; Zhang et al., 2022a); and (4) Human-in-the-loop mechanisms, which involve expert-driven refinement to narrow down potential detectors based on observed time series characteristics (Freeman & Beaver, 2019). Ensemble learning strategies that aggregate detector outputs using model variance as a selection signal (Jung et al., 2021) and methods that base selection on internal measures (Ma et al., 2021; Zhao et al., 2022) further extend these capabilities. Despite these advances, current evaluations are fragmented and often restricted to limited datasets. Among existing unsupervised selection methods, many have not been evaluated with time series anomaly detectors (Zhang et al., 2022b; Singh & Vanschoren, 2022; Zhao et al., 2021; Yu et al., 2022; Zhao et al., 2022; Ma et al., 2021), while others are tested on a limited set of detectors with few time series datasets (Navarro et al., 2023; Jung et al., 2021; Chatterjee et al., 2022; Goswami et al., 2021; Ying et al., 2020), leaving significant gaps in understanding model selection performance on multivariate time series data. A recent study offers useful insights into re-purposing time series classifiers for model selection in univariate anomaly detection (Sylligardos et al., 2023). However, benchmarking established model selection methods for multivariate time-series anomaly detection has remained largely underexplored. Our work aims to fill this gap by providing the first systematic benchmark that enables consistent comparison of selectors and analysis of their failure modes.

**Anomaly Detection Benchmarks.** Existing benchmarks for MTS-AD often fall short in terms of scale, diversity, and real-world representativeness. While several benchmarks have been proposed (Braei & Wagner, 2020; Dau et al., 2018; Paparrizos et al., 2022b; Laptev et al., 2015; Lavin & Ahmad, 2015; Lai et al., 2021; Schmidl et al., 2022; Wagner et al., 2023; Zhang et al., 2023; Liu & Paparrizos, 2024), only a handful include multivariate time series (Lai et al., 2021; Schmidl et al., 2022; Wagner et al., 2023; Zhang et al., 2023; Liu & Paparrizos, 2024), and even those predominantly focus on univariate time series or have very limited multivariate representation. Notably, none of these benchmarks support model selection evaluation, which is critical for deploying robust anomaly detectors in practice (Table 1). To address these limitations, mTSBench introduces the largest and most diverse multivariate time series collection from 19 publicly available sources, spanning 12 application domains. Most importantly, mTSBench is the *first* benchmark to comprehensively evaluate model selection strategies for MTS-AD, an important aspect that is underexplored in existing benchmarks.

## 3   mTSBench Benchmark

**Problem Definition.** A multivariate time series is a sequence of observations recorded over time, represented as $\mathcal{T} = \{(\mathbf{x}_1, t_1), (\mathbf{x}_2, t_2), \ldots, (\mathbf{x}_n, t_n)\}$, where $\mathbf{x}_i \in \mathbb{R}^d$ is a $d$-dimensional feature vector observed at timestamp $t_i$, and $n$ is the length of the time series (in practice, timestamps are often omitted). In supervised settings, each observation $\mathbf{x}_i$ in the time series can be associated with a binary anomaly label $y_i \in \{0, 1\}$, where $y_i = 1$ indicates an anomalous observation, and $y_i = 0$ indicates normal behavior. However, in practice, labels $\mathbf{y} = \{y_1, y_2, \ldots, y_n\}$ are often unavailable, necessitating the development of unsupervised methods for anomaly detection. Anomaly detection models $m_j : \mathbb{R}^d \to \mathbb{R}$ assign an anomaly score $s_i$ to each observation $\mathbf{x}_i$, *i.e.*, $s_i = m_j(\mathbf{x}_i)$, where a greater $s_i$ indicates a higher likelihood of $\mathbf{x}_i$ being anomalous. These scores are typically converted into binary predictions $\hat{y}_i \in \{0, 1\}$ using a threshold $\tau_j$, *i.e.*, $\hat{y}_i = \mathbb{I}(s_i > \tau_j)$.

There exists a large number of various anomaly detection algorithms for time series, yet no single one consistently outperforms others across all data distributions. This variability has driven the need for

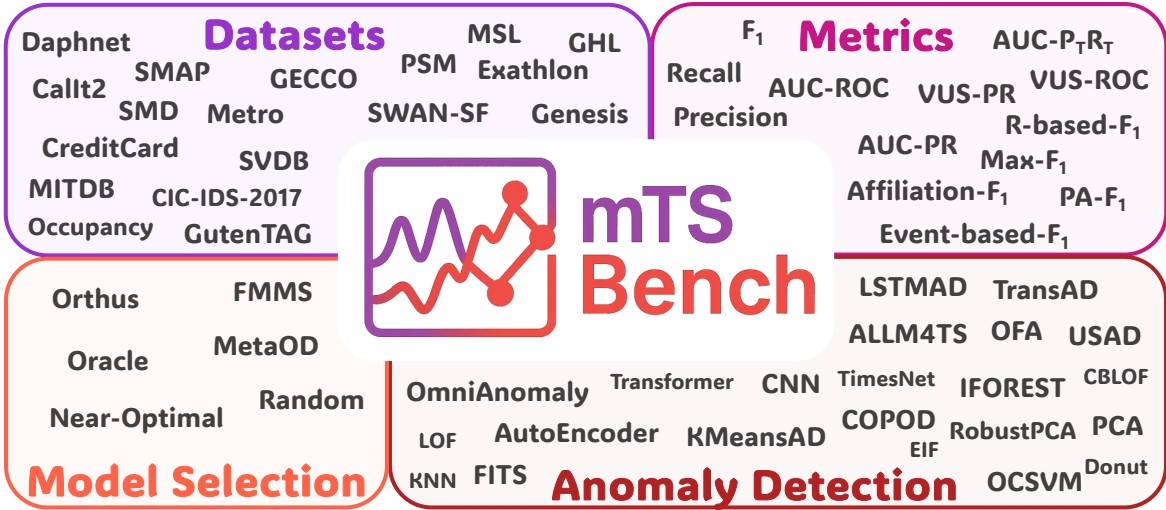

Figure 2: **mTSBench** **Overview.** mTSBench is the largest and most diverse benchmark for multivariate time series anomaly detection and model selection, spanning 19 multivariate time series datasets across various application domains and establishing a platform for robust anomaly detection and adaptive model selection in real-world multivariate contexts. mTSBench's comprehensive evaluation suite and diverse collection of state-of-the-art anomaly detectors, including statistical, deep learning, and LLM-based approaches, facilitates standardized comparison of model selection strategies.

unsupervised model selection methods to identify the most suitable anomaly detector for a specific dataset. Formally, let $\mathcal{M} = \{m_1, m_2, \ldots, m_k\}$ denote a set of $k$ anomaly detection models. The objective of unsupervised model selection for anomaly detection is to identify the best model $m^* \in \mathcal{M}$ for a given unlabeled test time series $\mathcal{T}_{\text{test}}$. In the absence of labels, unsupervised model selection relies on estimating the relative performance of models through a proxy performance function $\hat{P}(m_j, \mathcal{T}_{\text{test}})$, which approximates the performance of each model based on intrinsic characteristics of $\mathcal{T}_{\text{test}}$ and properties of the models in $\mathcal{M}$. The best model $m^*$ is selected as $m^* = \arg\max_{m_j \in \mathcal{M}} \hat{P}(m_j, \mathcal{T}_{\text{test}})$.

Model selection for MTS-AD has many applications across diverse industry sectors, including healthcare, cybersecurity, industrial monitoring, and finance, but remains underexplored due to the scarcity of comprehensive benchmarks that enable robust evaluation across multiple domains and realistic scenarios. mTSBench aims to fill this gap with a comprehensive suite of datasets, methods, and evaluation metrics.

### 3.1 mTSBench Overview

We introduce **mTSBench**, the largest and most diverse benchmark for MTS-AD and model selection to date. mTSBench consists of **344 multivariate time series** drawn from 19 publicly available sources and spans 12 application domains. Unlike existing benchmarks that primarily focus on univariate time series, mTSBench provides a rich representation of real-world scenarios by including both point-based and range-based anomalies, capturing complex temporal dependencies and cross-signal interactions. To enable robust evaluation, mTS-Bench integrates **24 anomaly detectors** spanning approaches based on reconstruction, prediction, statistics, and LLM, as well as **3 model selection methods** that leverage surrogate metrics, factorization machine, and meta-learning to identify the best anomaly detection method for a given time series. Finally, the evaluation suite in mTSBench includes **13 anomaly detection metrics** and **3 model selection metrics**, offering a comprehensive framework to benchmark efficacy and robustness of anomaly detection and model selection.

**Anomaly Detection Methods.** mTSBench has a diverse pool of open-source anomaly detectors, comprising 10 unsupervised and 14 semi-supervised methods, including currently existing LLMs designed for MTS-AD. The selected anomaly detection methods are summarized in Appendix A.

Unsupervised anomaly detection methods operate without the need for labeled training data and can be directly applied at test time to identify anomalies. In contrast, semi-supervised methods require training on

anomaly-free time series. mTSBench encompasses a wide variety of unsupervised methods: **CBLOF** (He et al., 2003), **COPOD** (Li et al., 2020), **EIF** (Hariri et al., 2019), **HBOS** (Goldstein & Dengel, 2012), **IForest** (Liu et al., 2008), **KMeansAD** (Yairi et al., 2001), **KNN** (Ramaswamy et al., 2000), **LOF** (Breunig et al., 2000), **PCA** (Aggarwal, 2017), and **RobustPCA** (Paffenroth et al., 2018), and semi-supervised methods: **LSTM-AD** (Malhotra et al., 2015), **AutoEncoder** (Sakurada & Yairi, 2014), **CNN** (Munir et al., 2018), **Donut** (Xu et al., 2018), **FITS** (Xu et al., 2023), **MCD** (Rousseeuw & Van Driessen, 1999), **OCSVM** (Schölkopf et al., 1999), **TimesNet** (Wu et al., 2023), **TranAD** (Tuli et al., 2022), **AnomalyTransformer** (Xu et al., 2022), **OmniAnomaly** (Su et al., 2019), and **USAD** (Audibert et al., 2020). mTSBench also includes recent foundation models designed for MTS-AD: **ALLM4TS** (Bian et al., 2024), and **OFA** (Zhou et al., 2023), which leverage large-scale pretraining to enable cross-domain generalization.

**Unsupervised Model Selection Methods.** To enable robust model selection across diverse multivariate time series, mTSBench includes three methods applicable to multivariate data. These include **MetaOD** (Navarro et al., 2023), a meta-learning method that employs Principal Component Analysis (PCA) to extract latent representations of a dataset and trains a random forest to map these representations to a precomputed performance matrix; **Orthus** (Zhao et al., 2021), a method that extracts univariate time-series meta-features and employs regression or random forests, depending on whether the test cases are unseen or similar to the training set; and **FMMS** (Zhang et al., 2022b), a factorization machine-based method that employs a regression model to directly relate meta-features to model performance.

In addition, we benchmark these model selection methods against three trivial baselines that establish performance bounds: **Oracle**, a baseline that consistently selects the best anomaly detection method for each time series (based on ground truth labels), serving as an upper bound; **Near-optimal**, a baseline that chooses the second-best anomaly detector for each time series, providing a practical reference that is more attainable than the theoretical maximum of the Oracle baseline; and **Random**, a baseline that selects an anomaly detector for each time series at random, offering a lower-bound estimate of model selection performance by disregarding dataset-specific characteristics. Additionally, we include **PCA**, a well-established classic machine learning method (Aggarwal, 2017), as a fixed-choice baseline; and an **Ensemble** baseline, where we compute the mean anomaly score across all 24 detectors at each time step to produce a single combined prediction (Sylligardos et al., 2023).

**Datasets.** Multivariate time series data in mTSBench span a wide range of application domains, including industrial process (GECCO (Rehbach et al., 2018), GHL (Filonov et al., 2016), Genesis (von Birgelen & Niggemann, 2018)), healthcare (Daphnet (Bachlin et al., 2009), MITDB (Goldberger et al., 2000), SVDB (Greenwald et al., 1990)), cybersecurity (CIC-IDS-2017 (Canadian Institute for Cybersecurity, 2017)), finance (CreditCard (Dal Pozzolo et al., 2018)), IT infrastructure (Exathlon (Jacob et al., 2021), PSM (Abdulaal et al., 2021), SMD (Su et al., 2019)), smart building (CalIt2 (Hutchins, 2006), Occupancy (Candanedo & Feldheim, 2016)), spacecraft telemetry (MSL, SMAP (Hundman et al., 2018)), *etc.* These time series vary significantly in scale, with lengths ranging from thousands to over half a million points, and dimensionalities ranging from $d = 3$ to $d = 73$, presenting a challenging and realistic testbed for evaluating MTS-AD and model selection strategies. The datasets also differ in their anomaly structures: some contain sparse point anomalies (*e.g.*, CreditCard and Occupancy), while others include complex or long-range anomalous sequences (*e.g.*, MITDB and CIC-IDS-2017). Each time series in mTSBench includes a clean training/test partition. To ensure consistency across datasets, we apply a unified data-splitting protocol: (1) if a dataset provides official train/test splits, we use them directly; (2) if anomaly labels are available but no official split is provided, we extract a long contiguous segment without labeled anomalies as the training set and use the remaining portion of the time series as the test set. Appendix B provides additional details on the dataset characteristics.

**Evaluation Metrics.** To provide a comprehensive evaluation, mTSBench reports 13 MTS-AD evaluation metrics that span both point-wise and range-based performance. Point-wise metrics include **Precision**, **Recall**, and **F$_1$**, which evaluate the ability of the selected model to accurately identify individual anomaly points while balancing false positives and false negatives. To capture broader detection quality, Area under the Receiver Operating Characteristics Curve (**AUC-ROC**), Area Under the Precision-Recall Curve (**AUC-PR**), and Area Under the range-based Precision, range-based Recall Curve (**AUC-P$_T$R$_T$**)are employed, assessing performance across varying thresholds (Tatbul et al., 2018). AUC-PR, in particular, is critical for detecting anomalies in imbalanced datasets where positive samples are sparse. Furthermore,

Table 2: **Model Selection Performance Comparison.** Entries are mean±std over 344 time series in mTSBench for the top-1 detector chosen by each method. Bold and ▨ denote best and second-best methods per metric. **Δ(%)** is the relative difference between mean values for the best method and Near-optimal.

| | Trivial Model Selection Baselines | | | | Unsupervised Model Selection Methods | | | | |
| Metric | Oracle | Near-optimal | Random | PCA | MetaOD | FMMS | Orthus | Ensemble | Δ(%) |
|---|---|---|---|---|---|---|---|---|---|
| **$F_1$** | 0.546 ± 0.263 | 0.514 ± 0.256 | 0.299 ± 0.252 | 0.252 ± 0.195 | 0.135 ± 0.135 | 0.222 ± 0.185 | **0.222 ± 0.200** | 0.189 ± 0.187 | **−56.80** |
| **Precision** | 0.414 ± 0.328 | 0.393 ± 0.319 | 0.194 ± 0.236 | 0.293 ± 0.305 | 0.189 ± 0.236 | 0.280 ± 0.299 | **0.297 ± 0.311** | 0.237 ± 0.262 | **−24.43** |
| **Recall** | 0.483 ± 0.298 | 0.458 ± 0.287 | 0.275 ± 0.313 | 0.319 ± 0.310 | 0.205 ± 0.210 | **0.307 ± 0.276** | 0.283 ± 0.271 | 0.300 ± 0.301 | **−32.97** |
| **Affiliation-$F_1$** | 0.879 ± 0.107 | 0.868 ± 0.101 | 0.785 ± 0.124 | 0.818 ± 0.108 | 0.777 ± 0.100 | **0.834 ± 0.112** | 0.814 ± 0.111 | 0.792 ± 0.107 | **−3.92** |
| **Event-based-$F_1$** | 0.699 ± 0.287 | 0.667 ± 0.283 | 0.427 ± 0.338 | 0.506 ± 0.329 | 0.377 ± 0.305 | **0.548 ± 0.333** | 0.500 ± 0.340 | 0.433 ± 0.330 | **−17.84** |
| **Max-$F_1$** | 0.546 ± 0.263 | 0.514 ± 0.256 | 0.299 ± 0.252 | 0.408 ± 0.268 | 0.292 ± 0.246 | 0.381 ± 0.263 | **0.386 ± 0.270** | 0.379 ± 0.270 | **−24.90** |
| **PA-$F_1$** | 0.825 ± 0.235 | 0.803 ± 0.226 | 0.705 ± 0.316 | 0.627 ± 0.320 | 0.658 ± 0.324 | **0.738 ± 0.285** | 0.647 ± 0.332 | 0.584 ± 0.332 | **−8.09** |
| **R-based-$F_1$** | 0.450 ± 0.237 | 0.430 ± 0.224 | 0.252 ± 0.190 | 0.410 ± 0.237 | 0.242 ± 0.194 | 0.366 ± 0.224 | **0.370 ± 0.236** | 0.328 ± 0.246 | **−13.95** |
| **AUC-$P_T R_T$** | 0.417 ± 0.276 | 0.400 ± 0.260 | 0.234 ± 0.228 | 0.322 ± 0.278 | 0.217 ± 0.218 | **0.318 ± 0.273** | 0.309 ± 0.272 | 0.278 ± 0.260 | **−20.50** |
| **AUC-PR** | 0.492 ± 0.296 | 0.455 ± 0.284 | 0.212 ± 0.222 | 0.327 ± 0.280 | 0.210 ± 0.231 | 0.304 ± 0.266 | **0.315 ± 0.279** | 0.287 ± 0.271 | **−30.77** |
| **AUC-ROC** | 0.811 ± 0.150 | 0.796 ± 0.148 | 0.613 ± 0.178 | 0.697 ± 0.219 | 0.605 ± 0.189 | 0.659 ± 0.207 | **0.672 ± 0.219** | 0.683 ± 0.214 | **−15.58** |
| **VUS-PR** | 0.524 ± 0.296 | 0.485 ± 0.290 | 0.236 ± 0.236 | 0.348 ± 0.290 | 0.229 ± 0.231 | 0.326 ± 0.277 | **0.333 ± 0.288** | 0.312 ± 0.279 | **−31.34** |
| **VUS-ROC** | 0.847 ± 0.134 | 0.832 ± 0.134 | 0.640 ± 0.188 | 0.721 ± 0.211 | 0.655 ± 0.179 | 0.698 ± 0.204 | **0.707 ± 0.212** | 0.714 ± 0.203 | **−15.02** |

range-based metrics Volume Under the Receiver Operating Characteristics Surface (**VUS-ROC**) and Volume Under the Precision-Recall Surface (**VUS-PR**) (Paparrizos et al., 2022a) generalize AUC by applying a tolerance buffer and continuous scoring over anomaly boundaries. Additional metrics include **PA-$F_1$** (Xu et al., 2018) that applies heuristic point adjustment, **Event-based-$F_1$** (Garg et al., 2021) that treats each anomaly segment as a single event, **Max-$F_1$** that reports the maximum $F_1$ score achievable across all decision thresholds, **R-based-$F_1$** (Tatbul et al., 2018) that captures structural properties such as existence, overlap, and cardinality, and **Affiliation-$F_1$** (Huet et al., 2022) that measures proximity between predicted and ground-truth intervals. Collectively, these metrics ensure mTSBench holistically evaluates anomaly detection methods across isolated, sequential, and sparse anomaly manifestations.

For model selection evaluation, mTSBench includes ranking metrics **Precision**@$k$, **Recall**@$k$, and Normalized Discounted Cumulative Gain (**NDCG**). Precision@$k$ evaluates the model selection method's ability to prioritize high-performing detectors within its top-$k$ recommendations, and refers to the proportion of selected detectors that are among the highest-performing ones for a given dataset. In contrast, Recall@$k$ quantifies the proportion of all high-performing detectors that are successfully retrieved within the top-$k$ recommendations. Finally, NDCG (or NDCG@$k$ for top-$k$ recommendations) evaluates how well model selection preserves the correct ranking order, taking into account the relevance and relative order of the selected anomaly detectors.

## 4 Experimental Results

### 4.1 Anomaly Detection Performance Across Evaluation Metrics

The results, visualized in Figure 3, reveal distinct performance clusters among detectors. PCA, despite its simplicity, demonstrates strong and stable performance, particularly in VUS-PR and AUC-ROC, highlighting its generalizability across diverse time series. OmniAnomaly, a deep generative model, performs comparably well across all metrics, including those sensitive to temporal structure such as AUC-$P_T R_T$, indicating its ability to model temporal dependencies. The higher median scores across VUS-PR and VUS-ROC suggest that these models excel not only in isolating anomalous regions but also in maintaining high relative rankings over various sensitivity levels, critical for long-tail anomaly distributions in multivariate time series. In contrast, methods such as KNN, Transformer, LOF, and HBOS occupy the lower end of the performance spectrum across all five metrics, with particularly poor AUC-PR and AUC-$P_T R_T$ scores. The low AUC-PR scores indicate struggling to detect anomalies in highly imbalanced datasets where true anomalies are sparse, while the poor AUC-$P_T R_T$ performance suggests an inability to effectively manage the precision-recall trade-off across thresholds. In contrast, methods like COPOD and RobustPCA, while similarly underperforming, display narrower distributions, suggesting stable but limited detection capacity.

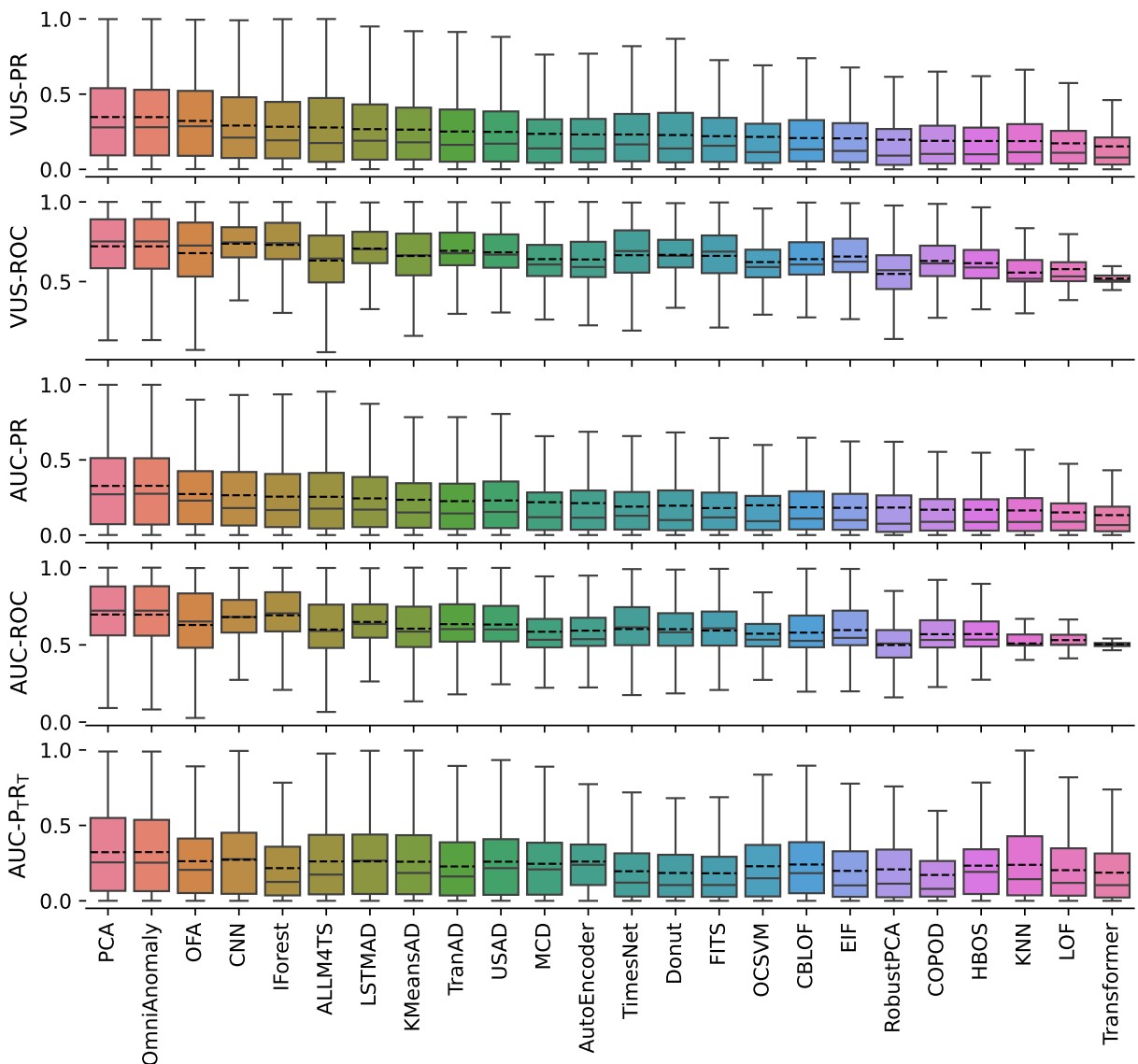

Figure 3: **Comparison of Anomaly Detection Methods Across Five Evaluation Metrics.** The boxplots illustrate the distribution of performance scores for each method evaluated over all **mTSBench** datasets, measured using VUS-PR, VUS-ROC, AUC-PR, AUC-ROC, and AUC-$P_T R_T$. Detectors are ordered by their average VUS-PR score. Boxes represent interquartile ranges, with solid lines indicating the median and dashed lines indicating the mean. In-depth analysis in §4.1.

In terms of LLM-based detectors, OFA consistently achieves higher median scores across VUS-PR, VUS-ROC, AUC-PR, AUC-ROC, and AUC-$P_T R_T$, with relatively compact interquartile ranges for AUC-ROC and AUC-$P_T R_T$ suggesting that it is less sensitive to distributional shifts. In contrast, ALLM4TS exhibits broader performance variability, particularly in VUS-PR and VUS-ROC, indicative of its sensitivity to dataset-specific noise. These observations suggest that both methods benefit from large-scale pretraining, where pretrained multimodal embeddings can capture richer temporal and semantic representations compared to traditional baselines. However, the observed variability in ALLM4TS and the performance gaps relative to more traditional baselines such as PCA suggest that significant improvements are still possible. For example, future work can explore modality-specific adapters, allowing LLM-based detectors to fine-tune their representations based on the temporal, spatial, and contextual properties of anomalies. The inherent instruction-following capability and generalization across diverse tasks of LLM-based methods can enable new human-in-the-loop

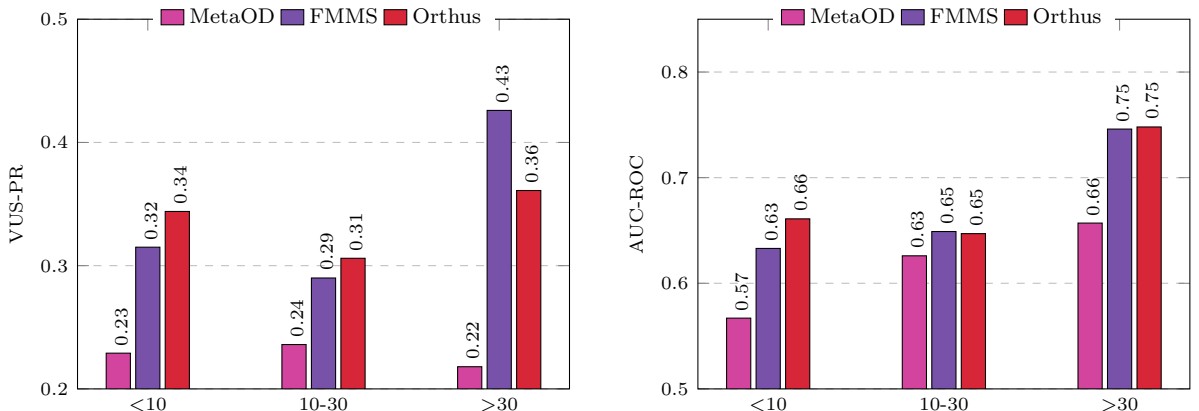

Figure 4: **Model Selection Performance Grouped by Time Series Dimensionality.** VUS-PR (left) and AUC-ROC (right) for three dimensionality groups ($<10$, $10$–$30$, $>30$). Discussion is in §4.2

anomaly detection and data analysis applications. To fully harness their potential, however, LLM-based detectors must demonstrate high adaptability to varying anomaly patterns and heterogeneous application domains. This necessitates robust model selection mechanisms that dynamically configure detection models to match the specific context and anomaly profile of incoming data.

## 4.2 Analysis of Model Selection Performance Across Evaluation Metrics

Table 2 presents the performance of the top-1 recommended anomaly detector for each model selection method across 13 evaluation metrics. FMMS and Orthus consistently outperform MetaOD across nearly all metrics, with Orthus achieving the best performance in 8 out of 13 metrics and FMMS leading in 5 metrics. MetaOD often performs on par with or worse than the Random baseline, suggesting limited transferability of its non-temporal meta-feature design to time-series settings. Orthus performs well in AUC-ROC and R-based-$F_1$, which indicates its strong capacity to minimize false positives and maintain high anomaly-ranking accuracy. Meanwhile, FMMS performs well in Recall and Event-based-$F_1$, suggesting it is more effective at capturing distributed and event-based anomalies. However, both Orthus and FMMS demonstrate significant gaps relative to the Near-optimal baseline, with a mean deficit of about 15% to 30% across key metrics. Their considerable underperformance in AUC-PR and VUS-PR, as compared to the Near-optimal baseline, means they select detectors that miss many true anomalies that best detectors would have caught. We also observe that PCA surpasses all three model selectors (MetaOD, FMMS, Orthus) on 9 out of 13 metrics, although its overall performance remains well below the Near-optimal and Oracle references. This further reinforces that existing model-selection approaches are far from reliable and that principled selection remains an open challenge in MTS-AD. The ensemble baseline performs even worse, trailing FMMS and MetaOD across all metrics, and achieving only comparable performance on AUC-ROC and VUS-ROC. These results highlight the limitations of the existing unsupervised model selection mechanisms, which currently lack the ability to adapt to varying characteristics of diverse time series data. Promising directions include meta-learning-based adaptation, continual learning, and self-supervised representations that could enable finer control and improved performance in non-stationary environments.

Additionally, Figure 4 shows performance grouped by time series dimensionality. FMMS and Orthus outperform MetaOD in both VUS-PR and AUC-ROC across all dimensionality groups. FMMS achieves its best performance for high dimensionalities ($d > 30$), while Orthus performs consistently well across all dimensionalities, highlighting its robustness across a wider range of settings. Beyond dimensionality, we also analyze selector performance with respect to anomaly ratio and anomaly sequence length. As shown in Table 3 (left), selector performance generally declines as the anomaly ratio increases. Both MetaOD and FMMS exhibit a monotonic decrease in AUC-ROC (MetaOD: $0.640 \rightarrow 0.545$; FMMS: $0.758 \rightarrow 0.608$), while Orthus peaks at moderate ratios (5–10%, 0.730) before dropping at $> 10\%$ (0.640). Although standard deviations are sizable and adjacent bins may overlap, the overall trend indicates that higher anomaly density

Table 3: **Model Selection Performance by Anomaly Characteristics.** Entries are mean±std of AUC-ROC across datasets grouped by anomaly ratio (left) and anomaly sequence length (right).    and    denote the best and second-best methods for the group, respectively. The best group is **bolded** for each selector. Discussion is in §4.2.

| | Anomaly Ratio | | | | Anomaly Sequence Length | | |
|---|---|---|---|---|---|---|---|
| **Group** | **MetaOD** | **FMMS** | **Orthus** | **Group** | **MetaOD** | **FMMS** | **Orthus** |
| <3% | **0.640 ± 0.206** | **0.758 ± 0.189** | 0.653 ± 0.269 | <50 | **0.671 ± 0.200** | **0.833 ± 0.148** | **0.848 ± 0.136** |
| 3–5% | 0.615 ± 0.168 | 0.676 ± 0.166 | 0.691 ± 0.187 | 50–100 | 0.542 ± 0.197 | 0.671 ± 0.156 | 0.705 ± 0.201 |
| 5–10% | 0.592 ± 0.148 | 0.641 ± 0.202 | **0.730 ± 0.185** | 100–200 | 0.619 ± 0.160 | 0.694 ± 0.181 | 0.641 ± 0.252 |
| >10% | 0.545 ± 0.151 | 0.608 ± 0.175 | 0.640 ± 0.184 | >200 | 0.576 ± 0.181 | 0.638 ± 0.203 | 0.666 ± 0.184 |

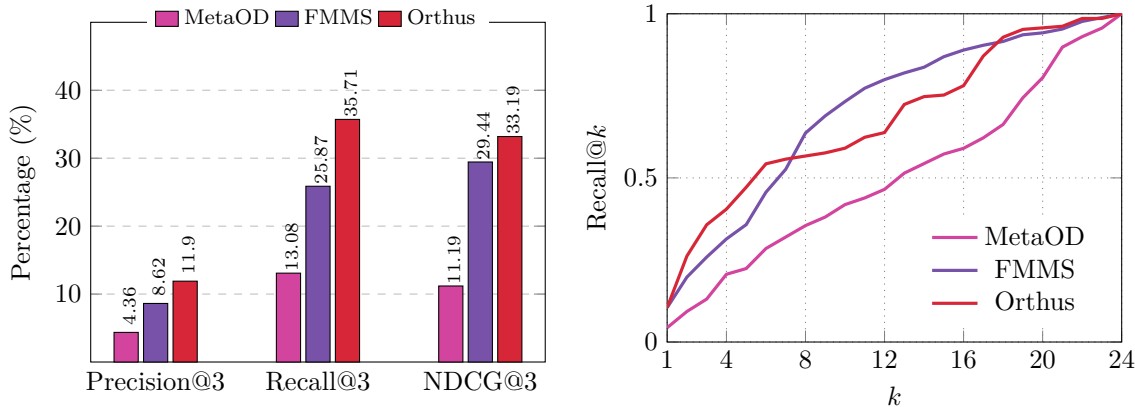

Figure 5: **Ranking Comparison of Model Selection Methods.** (Left) Precision@3, Recall@3, and NDCG@3 for each method. (Right) Recall@$k$ as a function of $k$. Discussion is in §4.3.

tends to make model selection more challenging across methods. In contrast, Table 3 (right) reveals that selector performance is more strongly affected by anomaly sequence length. All three methods perform better on datasets with shorter anomaly sequences, whereas performance degrades for longer segments. This reflects the difficulty of ranking detectors when anomalies span extended durations, where evaluation metrics (*e.g.*, AUC-ROC, Precision@k) are less sensitive to subtle localization errors. Shorter anomalies tend to produce more distinct and localized deviations from normal behavior, making it easier for detectors to identify them and for selectors to distinguish between well-performing and poorly performing detectors.

### 4.3 Comparison of Ranking Capability of Model Selection Methods

Figure 5 presents a comparative analysis of unsupervised model selection methods in terms of their ability to rank top-performing anomaly detectors. Top-3 results (Figure 5, left) show that Orthus achieves the highest scores for all three ranking metrics, indicating its superior ability to identify relevant anomaly detectors early in the ranked list. FMMS ranks second, while MetaOD is a distant third, performing only marginally better than random selection. However, even Orthus's NDCG@3 score of about 33% implies that its top-selected anomaly detector is often positioned far from the true top ranks, which highlights the need for continued progress toward more reliable adaptive model selection strategies across diverse time series contexts. In terms of overall ranking quality (Figure 5, right), Orthus and FMMS alternate in achieving better Recall@$k$ rates across broad ranges of $k$ values, while MetaOD remains consistently lower. We also evaluate selector performance across ranking depths by reporting Precision@$k$ and NDCG@$k$ as a function of $k$ (Figure 6). Orthus achieves the highest Precision@$k$ at small $k$, indicating a stronger ability to prioritize high-performing detectors in the very top positions. FMMS also performs competitively in this regime but exhibits a more gradual decline, reflecting stable ranking quality across intermediate depths. In contrast, MetaOD consistently lags behind other methods across all $k$, underscoring its limited effectiveness in the anomaly detection setting. For NDCG@$k$ (right panel), Orthus maintains an advantage at small to mid-$k$, while FMMS eventually matches or surpasses it for

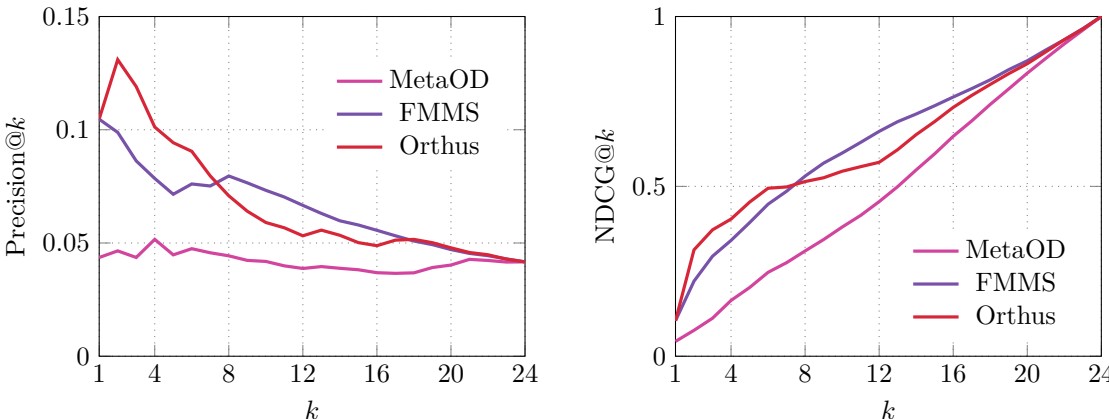

Figure 6: **Ranking Comparison of Model Selection Methods.** (Left) Precision@$k$ as a function of $k$. (Right) NDCG@$k$ as a function of $k$.

Table 4: **NDCG@5 by Anomaly Sequence Length.** Entries are mean±std of NDCG@5 grouped by anomaly sequence length. ■ and ■ denote best and second-best methods within each group, respectively. The best NDCG@5 is **bolded** for each selector. Discussion is in §4.3.

| Anomaly Seq. Length | MetaOD | FMMS | Orthus |
|---|---|---|---|
| <50 | $0.093 \pm 0.097$ | $0.364 \pm 0.257$ | $0.391 \pm 0.286$ |
| 50–100 | $0.000 \pm 0.000$ | $0.378 \pm 0.248$ | $0.519 \pm 0.286$ |
| 100–200 | $0.150 \pm 0.019$ | $\mathbf{0.448 \pm 0.264}$ | $\mathbf{0.519 \pm 0.197}$ |
| >200 | $\mathbf{0.214 \pm 0.000}$ | $0.357 \pm 0.227$ | $0.384 \pm 0.218$ |

larger values ($k > 6$). These findings suggest that current selectors are more effective at producing a shortlist of strong detectors than at reliably identifying a single best one, aligning with practical deployment needs.

To further examine robustness, we report NDCG@5 stratified by anomaly sequence length. As shown in Table 4, Orthus consistently outperforms FMMS and MetaOD across all groups. The gap is most significant for anomalies of length 50–100, where MetaOD fails to retrieve any of the top-5 detectors. Both FMMS and Orthus achieve their strongest performance on moderate-length anomalies (100–200), with NDCG@5 scores of 0.448 and 0.519, respectively. In contrast, MetaOD performs poorly on short anomalies (<50) and only marginally improves on long ones (>200), underscoring its limited robustness. These results reinforce the need for more adaptive selectors that can maintain ranking quality across varying anomaly characteristics.

To assess generalizability, we also compute domain-specific AUC-ROC scores for each model selector across four representative application domains (Table 5). Results indicate notable variation across domains. FMMS achieves the strongest performance in IT infrastructure (where it outperforms the other two selectors), but underperforms in healthcare. Orthus also achieves the strongest performance in IT infrastructure and shows good results in spacecraft telemetry and healthcare (outperforming the other two selectors in both these domains), while its weakest performance by a large margin is on industrial process data. MetaOD excels in industrial process (where it outperforms the other two selectors), but underperforms in spacecraft telemetry and IT infrastructure. These findings highlight that selector effectiveness is domain-dependent, reinforcing the need for domain-aware model selection strategies.

### 4.4 Inference Time Analysis of Model Selection Methods

Inference time is a critical consideration for practical deployment of anomaly detection systems, as the ability to rapidly detect and respond to anomalies can significantly impact the effectiveness of downstream decision-making processes in safety-critical applications. To assess the efficiency of each model selection

Table 5: **Domain-level Model Selection Performance.** Entries are mean±std of AUC-ROC across four representative application domains. ▮ and ▮ denote the best and second-best methods for the domain, respectively. The best group is **bolded** for each selector. Results show clear variability across domains, highlighting the importance of domain-aware selection strategies.

| Domain | MetaOD | FMMS | Orthus |
|---|---|---|---|
| Healthcare | $0.654 \pm 0.122$ | $0.617 \pm 0.161$ | $0.661 \pm 0.168$ |
| IT Infrastructure | $0.636 \pm 0.221$ | $\mathbf{0.790 \pm 0.152}$ | $\mathbf{0.770 \pm 0.176}$ |
| Industrial Process | $\mathbf{0.792 \pm 0.198}$ | $0.774 \pm 0.160$ | $0.344 \pm 0.343$ |
| Spacecraft Telemetry | $0.514 \pm 0.247$ | $0.564 \pm 0.223$ | $0.703 \pm 0.224$ |

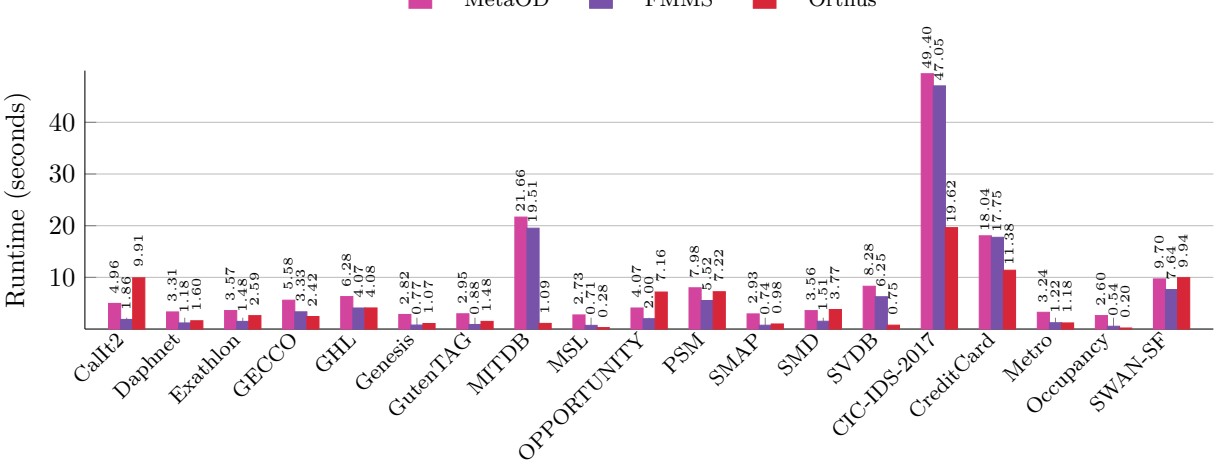

Figure 7: **Average Inference Time of Model Selection Methods.** Discussion is in §4.4.

method, we analyze their average inference times on mTSBench. Figure 7 shows that FMMS is generally the fastest selector across most datasets, with runtimes typically under 2 s, except for larger datasets like CIC-IDS-2017 (47.05 s), MITDB (19.51 s), and CREDITCARD (17.75 s), where other selectors also experience increased latency. Orthus consistently achieves a strong balance between inference efficiency and detection accuracy, with typically moderate runtimes (*e.g.*, 9.94 s on SWAN-SF and 7.22 s on PSM) and even best efficiency on larger datasets (*e.g.*, 1.09 s on MITDB), while maintaining strong performance. MetaOD exhibits faster inference in certain cases (*e.g.*, GHL, GENESIS), but shows higher variability and scales poorly on larger datasets, *e.g.*, MITDB (21.66 s) and CIC-IDS-2017 (49.40 s). These findings show the challenge of achieving both fast and high-quality recommendations as time series grow in length and complexity.

## 5 Practical Takeaways

Across datasets in mTSBench, the evaluated unsupervised selectors consistently underperform optimal or oracle baselines, indicating that effective model selection remains challenging in MTS-AD. This gap reflects several recurring limitations observed in our evaluation. For example, selectors rely primarily on coarse, dataset-level meta-features, which provide limited information about temporal structure, long-range dependencies, or cross-signal interactions that are critical for distinguishing detector behavior across heterogeneous time series. Moreover, detector performance varies substantially with both time-series properties (*e.g.*, dimensionality, noise characteristics, and cross-channel coupling) and anomaly characteristics (*e.g.*, density, duration, and temporal dispersion), yet these factors are not explicitly represented in existing selector features. As a result, selectors trained on aggregate statistics as meta-features exhibit limited ability to adapt to dataset-specific characteristics, contributing to the observed gap relative to oracle performance in mTSBench.

Our results also suggest several considerations for interpreting unsupervised model selection outcomes. The persistent gap between selector performance and oracle baselines indicates that identifying a single best detector is difficult under dataset heterogeneity and limited prior information. Accordingly, top-1 selector outputs should be interpreted with caution and may be better viewed as coarse rankings of candidate detectors rather than definitive choices. In such settings, relying on well-established detectors or considering a small set of high-ranked candidates can provide more stable behavior when domain knowledge is limited.

## 6 Conclusion

We introduce **mTSBench**, the largest, most comprehensive benchmark for MTS-AD and model selection. Through systematic evaluation across 344 labeled time series from diverse application domains, mTSBench highlights the substantial variability in anomaly detection performance and the critical need for robust model selection strategies. Empirical results demonstrate that existing unsupervised model selection methods, while promising, fall significantly short of optimal performance within mTSBench, exposing critical gaps in handling complex temporal dependencies and cross-signal interactions. To address these limitations, mTSBench provides a unified evaluation suite to facilitate reproducible research and accelerate progress in robust anomaly detection and adaptive model selection, enabling more resilient multivariate time series analysis across domains like healthcare, industrial monitoring, and cybersecurity. These findings motivate new research directions in context-aware model selection, adaptive selectors that dynamically respond to temporal shifts, and integration with foundation models to enhance cross-domain generalization and robustness.

### Broader Impact Statement

This work introduces a standardized benchmark for model selection in MTS-AD, encouraging the development of more robust, adaptive, and generalizable anomaly detection systems. The societal impact spans various high-stakes domains, including healthcare, industrial monitoring, cybersecurity, and financial systems, where reliable anomaly detection can enhance safety, efficiency, and decision-making. Model selection in MTS-AD has the potential to improve early warning mechanisms, reduce downtime in safety-critical systems, and support human decision-making in complex temporal environments. However, we recognize that deploying model selection methods without robust validation could lead to unintended consequences, such as false alarms or missed critical events. While our benchmark highlights generalization gaps in existing methods, it is important that future research also considers fairness, transparency, and robustness to data distribution shifts, particularly in sensitive domains. It is also important that future research considers computational efficiency and energy requirements, especially as LLM-based and deep learning selectors become more prevalent. Balancing accuracy, efficiency, and resource usage will be essential for responsible and scalable deployment of model selection methods. By open-sourcing mTSBench, we aim to facilitate access to rigorous evaluation tools, foster reproducibility, and accelerate research in time series analysis. We anticipate this work will support practitioners in deploying more effective and trustworthy anomaly detection pipelines, while also inspiring new methods that adapt to complex, real-world data.

### Acknowledgments

This work was supported by the U.S. Department of Energy, National Nuclear Security Administration, Office of Defense Nuclear Nonproliferation Research and Development, and by the Laboratory Directed Research and Development program at Sandia National Laboratories. Sandia National Laboratories is a multi-mission laboratory managed and operated by National Technology & Engineering Solutions of Sandia, LLC (NTESS), a wholly owned subsidiary of Honeywell International Inc., for the U.S. Department of Energy's National Nuclear Security Administration (DOE/NNSA) under contract DE-NA0003525. This written work is authored by an employee of NTESS. The employee, not NTESS, owns the right, title, and interest in and to the written work and is responsible for its contents. Any subjective views or opinions that might be expressed in the written work do not necessarily represent the views of the U.S. Government. The publisher acknowledges that the U.S. Government retains a non-exclusive, paid-up, irrevocable, worldwide license to publish or reproduce the published form of this written work or allow others to do so, for U.S. Government purposes. The DOE will provide public access to results of federally sponsored research in accordance with the DOE Public Access Plan.

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

# mTSBench: Appendices

**Table of Contents:**

## A  Anomaly Detectors

mTSBench includes 24 anomaly detectors applicable to multivariate time series (MTS), as shown in Table A.1, spanning four major areas: outlier detection, classic machine learning (ML), deep learning, and large language models (LLMs). These can be grouped into six method families based on their underlying detection strategies. **Distance-based methods** identify anomalies based on their proximity to cluster centers or nearest neighbors. **Distribution-based methods** model statistical properties and detect deviations from expected distributions. **Reconstruction-based methods** learn compact representations of the input and flag large reconstruction errors as anomalies. **Tree-based methods** rely on recursive partitioning of the feature space to isolate anomalies, leveraging the principle that anomalies are easier to separate due to their sparsity. **Forecasting-based methods** predict future time-series values from prior data and detect anomalies as significant deviations from the forecast. **Foundation model-based methods** utilize pre-trained LLMs to perform anomaly detection.

Table A.1: **Overview of Anomaly Detection Methods in mTSBench**, encompassing a diverse range of techniques, including foundation models, distance-based, forecasting-based, and reconstruction-based approaches, ensuring a comprehensive representation of different methodological paradigms.

| Learning | Anomaly Detection Method | Area | Method Family |
|---|---|---|---|
| **Unsupervised** | CBLOF (He et al., 2003) | Outlier Detection | Distance |
| | COPOD (Li et al., 2020) | Outlier Detection | Distribution |
| | EIF (Hariri et al., 2019) | Classic ML | Tree |
| | HBOS (Goldstein & Dengel, 2012) | Classic ML | Distance |
| | IForest (Liu et al., 2008) | Outlier Detection | Tree |
| | KMeansAD (Yairi et al., 2001) | Classic ML | Distance |
| | KNN (Ramaswamy et al., 2000) | Classic ML | Distance |
| | LOF (Breunig et al., 2000) | Outlier Detection | Distance |
| | PCA (Aggarwal, 2017) | Classic ML | Reconstruction |
| | RobustPCA (Paffenroth et al., 2018) | Classic ML | Reconstruction |
| **Semi-supervised** | ALLM4TS (Bian et al., 2024) | LLM | Foundation Model |
| | Transformer (Xu et al., 2022) | Deep Learning | Reconstruction |
| | AutoEncoder (Sakurada & Yairi, 2014) | Deep Learning | Reconstruction |
| | CNN (Munir et al., 2018) | Deep Learning | Forecasting |
| | Donut (Xu et al., 2018) | Deep Learning | Reconstruction |
| | FITS (Xu et al., 2023) | Deep Learning | Forecasting |
| | LSTMAD (Malhotra et al., 2015) | Deep Learning | Forecasting |
| | MCD (Rousseeuw & Van Driessen, 1999) | Classic ML | Reconstruction |
| | OCSVM (Schölkopf et al., 1999) | Outlier Detection | Distribution |
| | OFA (Zhou et al., 2023) | LLM | Foundation Model |
| | OmniAnomaly (Su et al., 2019) | Deep Learning | Reconstruction |
| | TimesNet (Wu et al., 2023) | Deep Learning | Forecasting |
| | TranAD (Tuli et al., 2022) | Deep Learning | Forecasting |
| | USAD (Audibert et al., 2020) | Deep Learning | Reconstruction |

Table B.1: **Overview of the 19 MTS datasets used in mTSBench.** For each dataset, we record its domain, number of time series (#TS), dimensionality (#Dim), average time series length, number of anomalous points (#AnomPts) and anomalous sequences (#AnomSeqs) per time series, and license.

| Dataset | Domain | #TS | #Dim | Length | #AnomPts | #AnomSeqs | License |
|---|---|---|---|---|---|---|---|
| **CIC-IDS-2017** (Canadian Institute for Cybersecurity, 2017) | Cybersecurity | 4 | 73 | >100K | 0–8656 | 0–2546 | Citation Required |
| **CalIt2** (Hutchins, 2006) | Smart Building | 1 | 2 | >5K | 0 | 21 | CC BY 4.0 |
| **CreditCard** (Dal Pozzolo et al., 2018) | Finance / Fraud Detection | 1 | 29 | >100K | 219 | 10 | Citation Required |
| **Daphnet** (Bachlin et al., 2009) | Healthcare | 26 | 9 | >50K | 0 | 1–16 | CC BY 4.0 |
| **Exathlon** (Jacob et al., 2021) | IT Infrastructure | 30 | 20 | >50K | 0–4 | 0–6 | Apache 2.0 |
| **GECCO** (Rehbach et al., 2018) | Industrial Process | 1 | 9 | >50K | 0 | 37 | Citation Required |
| **GHL** (Filonov et al., 2016) | Industrial Process | 14 | 16 | >100K | 0 | 1–4 | Contact Authors |
| **Genesis** (von Birgelen & Niggemann, 2018) | Industrial Process | 1 | 18 | >5K | 0 | 2 | CC BY-NC-SA 4.0 |
| **GutenTAG** (Schmidl et al., 2022) | Synthetic Benchmark | 30 | 20 | >10K | 0 | 1–3 | MIT |
| **MITDB** (Goldberger et al., 2000) | Healthcare | 47 | 2 | >500K | 0 | 1–720 | ODC-By v1.0 |
| **MSL** (Hundman et al., 2018) | Spacecraft Telemetry | 26 | 55 | >5K | 0 | 1–3 | BSD 3-Clause |
| **Metro** (Helwig et al., 2015) | Transportation | 1 | 5 | >10K | 20 | 5 | CC BY 4.0 |
| **OPPORTUNITY** (Roggen et al., 2010) | Human Activity Recognition | 13 | 32 | >25K | 0 | 1 | CC BY 4.0 |
| **Occupancy** (Candanedo & Feldheim, 2016) | Smart Building | 2 | 5 | >5K | 1–3 | 9–13 | CC BY 4.0 |
| **PSM** (Abdulaal et al., 2021) | IT Infrastructure | 1 | 26 | >50K | 0 | 39 | CC BY 4.0 |
| **SMAP** (Hundman et al., 2018) | Spacecraft Telemetry | 48 | 25 | >5K | 0 | 1–3 | BSD 3-Clause |
| **SMD** (Su et al., 2019) | IT Infrastructure | 18 | 38 | >10K | 0 | 4–24 | MIT |
| **SVDB** (Greenwald et al., 1990) | Healthcare | 78 | 2 | >100K | 0 | 2–678 | ODC-By v1.0 |
| **SWAN-SF** (Angryk et al., 2020) | Astrophysics | 1 | 38 | >50K | 5233 | 1382 | MIT |

# B  Time Series Datasets

As summarized in Table B.1, mTSBench contains 344 multivariate time series from 19 datasets, covering a range of domains, anomaly types, and time series characteristics to ensure a diverse and representative benchmark. In Figure B.1, we visualize four example time series to illustrate the variability in sequence length and anomaly patterns. For example, some time series (*e.g.*, METRO) contain short and rare anomalies, while others (*e.g.*, CALIT2 and OCCUPANCY) exhibit longer or more frequent anomalous segments. Brief descriptions of the 19 datasets are provided below.

**CIC-IDS-2017** (Canadian Institute for Cybersecurity, 2017) contains network traffic data, including benign behavior and a wide range of attack scenarios such as DDoS, brute-force, and infiltration. This dataset provides labeled flow-based features extracted from raw PCAP files, enabling evaluation of intrusion detection systems.

**CalIt2** (Ihler et al., 2006) consists of data recording the number of people entering and exiting the main door of the CalIt2 building at UCI over 15 weeks, 48 time slices per day. The goal is to detect events, such as conferences, indicated by unusually high people counts during specific days or time periods.

**CreditCard** (Dal Pozzolo et al., 2018) contains anonymized variables derived from a subset of 284,807 online credit card transactions (including 492 fraudulent transactions) that occurred during two days in September 2013. Due to extreme class imbalance, this dataset is well-suited for evaluating anomaly detection methods in high-stakes, imbalanced classification scenarios.

**Daphnet** (Bachlin et al., 2009) consists of annotated readings from three acceleration sensors placed on the hip and leg of Parkinson's disease patients who experience freezing of gait (FoG) during walking tasks, such as straight line walking and walking with numerous turns.

**Exathlon** (Jacob et al., 2021) is a benchmark for explainable anomaly detection in high-dimensional time series, based on real Apache Spark stream processing traces. Executions were intentionally disturbed by introducing six types of anomalous events (bursty input, bursty input until crash, stalled input, CPU contention, driver failure, and executor failure).

**GECCO** (Rehbach et al., 2018) was originally introduced in the GECCO Challenge 2017. This dataset contains sensor readings from water treatment plants and distribution systems, capturing parameters such as water quality, flow rates, and pressure.

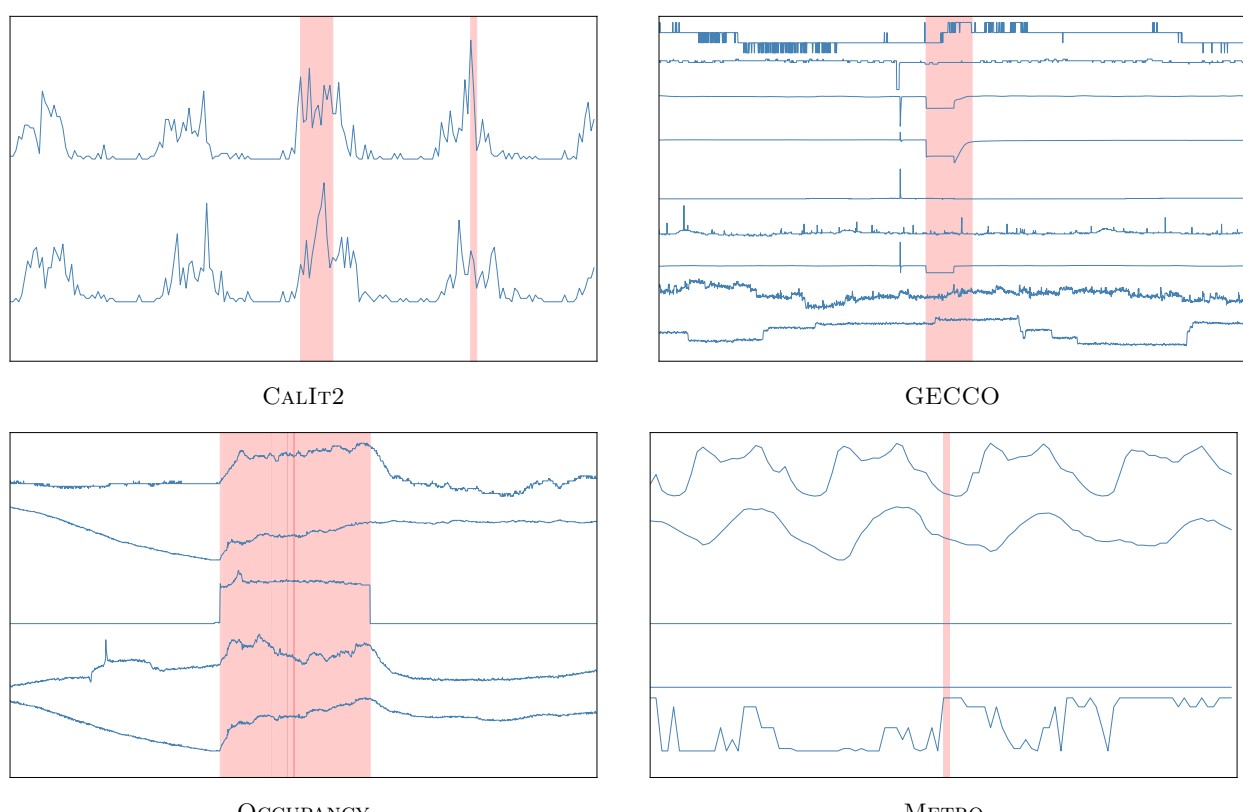

Figure B.1: **Examples of MTS Segments in mTSBench** spanning diverse domains and temporal patterns. From top left to bottom right: CALIT2 (smart building sensors), GECCO (water quality monitoring), OCCUPANCY (smart building sensors), and METRO (transportation systems). Shaded red regions indicate ground-truth anomalies, highlighting the variability in anomaly characteristics across datasets.

**GHL** (Filonov et al., 2016) records the operational status of three reservoirs, including variables such as temperature and water level. Anomalies correspond to shifts in maximum temperature or pump frequency.

**Genesis** (von Birgelen & Niggemann, 2018) contains sensor readings collected from a portable pick-and-place demonstrator, a cyber-physical system that sorts two different materials (conductive and non-conductive) from a magazine into their corresponding target locations. The dataset contains five primary continuous signals, thirteen discrete signals, and a single Unix timestamp. Two out of 42 production cycles contain anomalous behavior.

**GutenTAG** is a synthetic anomaly detection benchmark dataset. We adapted the implementation from (Wenig et al., 2022) to create a customized version of the original GutenTAG dataset, consisting of 30 multivariate time series, each comprising 20 dimensions. The dataset is partitioned into training, validation, and testing sets with shapes $(1000, 20)$, $(4000, 20)$, and $(10000, 20)$, respectively. The underlying signals are derived from six base oscillation types: CBF, ECG, Random Mode Jump, Sawtooth, Dirichlet, and MLS. Each base type is used in 100 dimensions, resulting in a total of 600 unique dimensions across the dataset. Anomalies are drawn from 10 types: *amplitude*, *extremum*, *frequency*, *mean*, *pattern*, *pattern_shift*, *platform*, *trend*, *variance*, and *mode-correlation* and injected into a randomly chosen subset of 1, 2, 5, 10, or all 20 dimensions for each time series. Each anomaly-affected dimension contains 1 to 3 anomalies of length 50 to 200 points, located randomly between 10% and 90% of the time series. The type of anomaly is selected randomly from the set of types compatible with the base oscillation in that dimension.

**MITDB** (Goldberger et al., 2000) consists of 48 half-hour excerpts of two-channel ambulatory ECG recordings collected from 47 individuals by the BIH Arrhythmia Laboratory during the years 1975 to 1979.

Note that the time series in this dataset contain trivial anomalies that are similar to one another, and the performance of detectors on MITDB should be interpreted with caution.

**MSL** (Hundman et al., 2018) originates from NASA's Mars Science Laboratory mission, and includes telemetry data from the Curiosity rover.

**Metro** (Helwig et al., 2015) contains hourly traffic volume data from a highway in Minneapolis–St Paul, collected by a single loop detector over several years, and includes features such as weather conditions, time-based attributes, and traffic volume counts.

**OPPORTUNITY** (Roggen et al., 2010) is a benchmark dataset for human activity recognition, encompassing tasks such as classification, segmentation, sensor fusion, and feature extraction. The dataset consists of motion sensor recordings collected as users performed routine daily activities.

**Occupancy** (Candanedo & Feldheim, 2016) consists of indoor environmental sensor readings, including temperature, humidity, light, and $CO_2$ levels, collected from a monitored room to predict occupancy status. The data is labeled and time-stamped, supporting tasks such as binary classification, energy efficiency modeling, and real-time occupancy prediction.

**PSM** (Abdulaal et al., 2021) contains data collected over 21 weeks from multiple application server nodes at eBay. It includes 26 variables that describe server machine metrics such as CPU utilization and memory.

**SMAP** (Hundman et al., 2018) originates from NASA's Soil Moisture Active Passive satellite mission, and contains telemetry data collected from the satellite's sensors, capturing various operational aspects of the satellite.

**SMD** (Su et al., 2019) is composed of time series data collected over a five-week period from server machines in a data center. This dataset captures metrics such as CPU usage, memory consumption, disk I/O, and network traffic, making it a representative dataset for monitoring IT infrastructure systems.

**SVDB** (Greenwald et al., 1990) contains 78 half-hour ECG recordings featuring a combination of supraventricular and ventricular ectopic beats within a normal sinus rhythm. Similar to MITDB, detector performance on this dataset should be interpreted with caution.

**SWAN-SF** (Angryk et al., 2020) comprises multivariate time series data derived from solar photospheric vector magnetograms in the SHARP data series, designed for space weather analytics and solar flare prediction.

## C  Time Series Data Quality Case Studies

Dataset quality plays a critical role in the reliable evaluation of anomaly detection methods. Inaccurate or inconsistent labels, such as normal points mislabeled as anomalies or vice versa, can systematically distort evaluation results, penalizing detectors that correctly identify true anomalies while favoring methods that produce random or misaligned predictions. Consequently, detector performance measured on low-quality datasets may not reflect the true capabilities of the underlying models. To mitigate the influence of such noise, mTSBench includes a large and diverse collection of 344 labeled time series, reducing the impact of individual low-quality instances and enabling more robust, aggregate performance analysis across datasets.

In addition, to better understand how time-series characteristics and labeling quality affect anomaly detection, Figure C.1 contrasts datasets with ambiguous versus well-justified anomaly annotations. In EXATHLON and GHL, the labeled anomalous segments exhibit limited visual deviation from surrounding behavior, making it difficult to identify clear temporal or cross-dimensional patterns that justify their annotation. Such ambiguity suggests the presence of label noise or dataset-specific annotation criteria, which may penalize detectors that capture meaningful but unlabeled deviations. In contrast, GUTENTAG and DAPHNET exhibit anomalies that align with clear and interpretable signal changes visible in the plots. In GUTENTAG, the anomalous interval in the ninth dimension (counting from top) shows a marked increase in oscillation frequency relative to both preceding and subsequent segments. Similarly, in DAPHNET, the labeled anomalies correspond to pronounced changes in frequency and fluctuation patterns compared to normal behavior in multiple dimensions. These examples illustrate how differences in label quality and anomaly salience can substantially influence detector performance and underscore the importance of careful dataset curation when benchmarking anomaly detection and model selection methods.

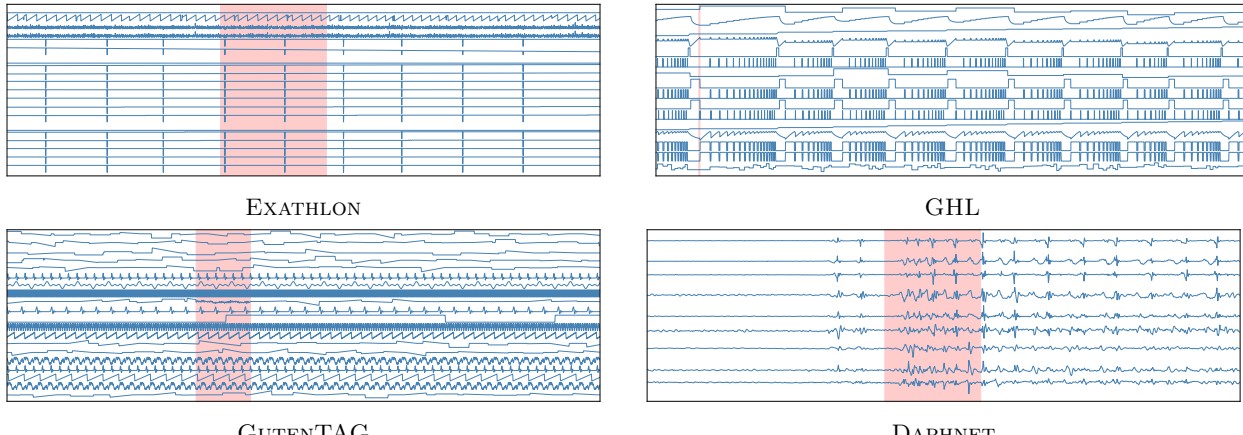

Figure C.1: **Representative time series from mTSBench** highlighting differences in label quality (red shaded regions denote ground-truth anomalies). In EXATHLON and GHL, anomalous segments lack clear visual justification. In contrast, anomalies in GUTENTAG (ninth dimension) and DAPHNET correspond to pronounced changes in frequency and fluctuation patterns.

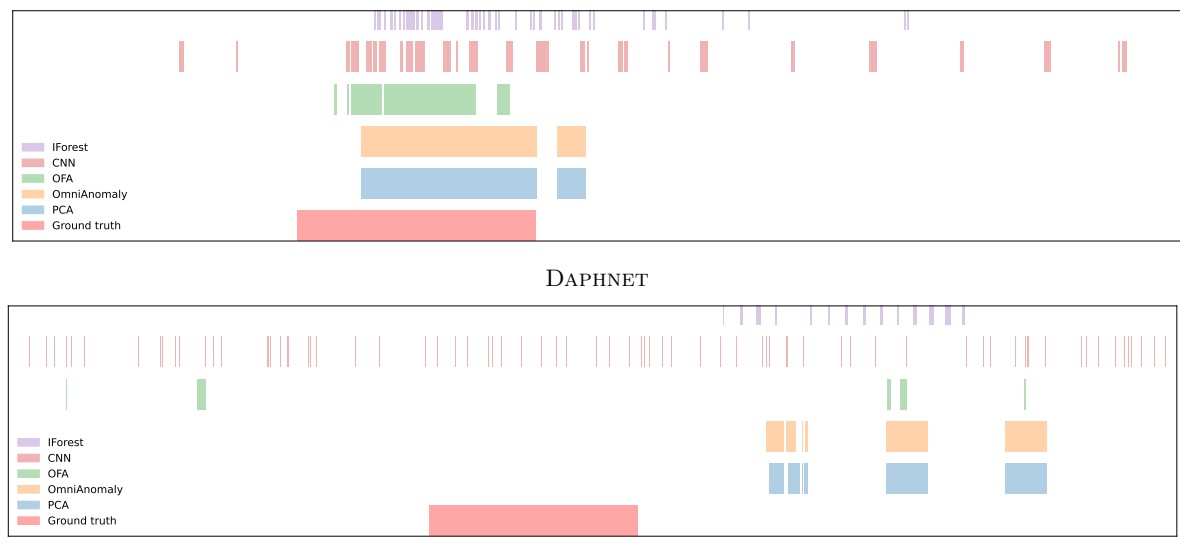

Figure C.2: **Ground Truth versus Predicted Anomalies**. In DAPHNET (top), predictions from all five detectors overlap with the ground-truth anomaly interval. In EXATHLON (bottom), only the CNN detector exhibits partial overlap with the ground-truth anomaly.

We further illustrate the impact of labeling quality by comparing ground-truth anomalies with predictions from five representative detectors, PCA, OmniAnomaly, OFA, CNN, and IForest, in Figure C.2. For DAPHNET, where the labeled anomalous interval aligns with clear signal changes in the plot, predictions from all five detectors overlap with the ground-truth anomaly. In addition, PCA, OmniAnomaly, and OFA produce predictions that closely match the annotated interval, whereas CNN and IForest generate anomalous regions that only partially overlap with the labeled boundaries and extend well outside the ground-truth anomaly. In contrast, for EXATHLON, where anomaly annotations are visually ambiguous, only CNN shows partial overlap with the labeled anomaly. However, this overlap is difficult to interpret, as CNN also predicts anomalies across much of the window, making it challenging to distinguish meaningful detections from spurious ones.

These examples illustrate how poor or ambiguous labeling can obscure differences between stronger and weaker detectors, limiting the interpretability of anomaly detection results. Improving label quality is itself

challenging, as the datasets span diverse domains and often require expert knowledge for reliable reannotation. An alternative approach is to examine agreement across multiple detectors: when several methods identify similar anomalous regions, as in DAPHNET, annotations are more likely to reflect true underlying irregularities. However, such an agreement is neither guaranteed nor sufficient, as it may reflect shared inductive biases among detectors or favor specific anomaly characteristics. This motivates our use of mTSBench, which aggregates results over 344 time series, reducing the influence of a small number of unknown low-quality datasets and enabling more robust, large-scale evaluation.

## D   Model Selection Methods for Time Series Anomaly Detection

mTSBench contains three unsupervised model selection methods for MTS-AD, described below.

**MetaOD** (Zhao et al., 2021) is a meta-learning framework that estimates a model's performance on a new dataset by leveraging knowledge from its prior performance on training datasets. MetaOD operates in two phases: offline training and online model selection. During offline training, it evaluates the models on all datasets, constructing a performance matrix that records each model's performance on each dataset. MetaOD extracts meta-features from the datasets, capturing intrinsic properties such as statistical summaries and landmark features. To reduce dimensionality, the extracted meta-features are processed through Principal Component Analysis (PCA), producing latent representations that initialize a dataset matrix. Simultaneously, a model matrix is initialized with values from a normal distribution, and matrix factorization is applied to approximate the relationship between datasets and models. This factorization is optimized using a rank-discounted cumulative gain (DCG) objective to uncover latent interactions. After matrix factorization, a regression model is trained to map the meta-features to the optimized latent representations of datasets. In the online phase, when a new dataset is introduced, its meta-features are extracted and reduced through PCA to generate latent representations. These representations are then passed through the trained regression model to compute the dataset's position in the latent space. By combining this latent representation with the precomputed model matrix, MetaOD estimates the performance of each model on the new dataset, and selects the model with the highest estimated performance score as the most suitable for the task.

**FMMS** (Zhang et al., 2022b) (Factorization Machine-based Unsupervised Model Selection) employs factorization to transfer model performance on prior known datasets into a second-order regression function that describes the relationship between the dataset's meta-features and its performance matrix. FMMS incorporates the meta-feature matrix, the interactions between features, and regression parameters to predict model performance. The regression function is optimized using a cosine distance loss to ensure accurate predictions. During inference, meta-features of the test dataset are passed through the trained regression function to estimate model performance. The model with the highest predicted performance is selected.

**Orthus** (Navarro et al., 2023) extracts a set of 22 univariate time series meta-features, summarized using five statistics: `{min, first quartile, mean, third quartile, max}`. This results in a total of 110 meta-feature values. Orthus splits the model selection problem into two scenarios: when the test data is novel and when the test data is similar to the training data. To determine the scenario, test data is projected in meta-feature space using Uniform Manifold Approximation and Projection (UMAP). If the test data belongs to a cluster that does not contain any points from training data, it is considered novel, and recommendations are estimated via URegression, which applies Singular Value Decomposition to the performance matrix and fits a Random Forest regressor to predict the decomposed components from meta-features. During inference, the test data's meta-features are used to estimate performance across models, selecting the one with the highest predicted performance. Otherwise, if the test data is similar to the training data, UMAP is applied to the performance matrix to form clusters, each with its own regressor. The test meta-features are evaluated across all cluster-specific regressors, and the top-performing model is selected.

## E   Impact of MTS Anomaly Length on Detection Performance

To investigate the impact of anomaly length on detection performance, we evaluate the models across time series with extreme anomaly durations. Specifically, we focus on two cases: (1) long anomalies, where the average anomaly length exceeds 2500 data points, and (2) short anomalies, where the average anomaly

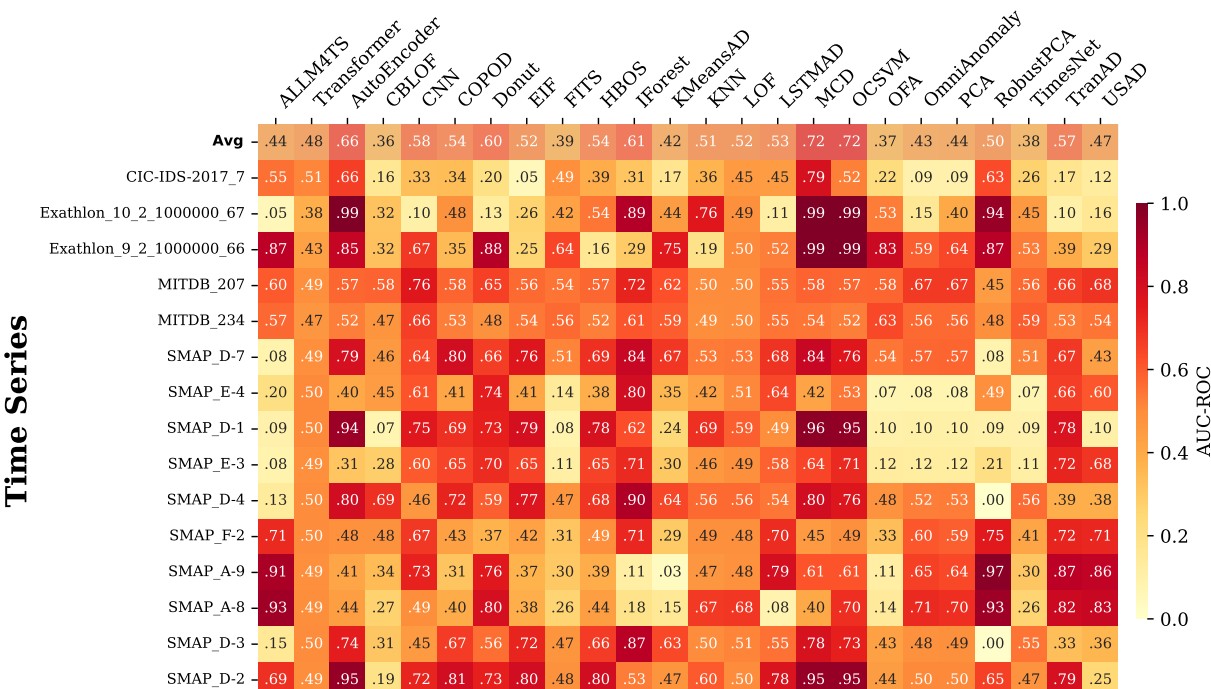

Figure E.1: **AUC-ROC (↑) of Anomaly Detectors ($x$-axis) Evaluated on MTS with Extremely Long Anomalies in mTSBench ($y$-axis).** Top row presents the average AUC-ROC over these MTS.

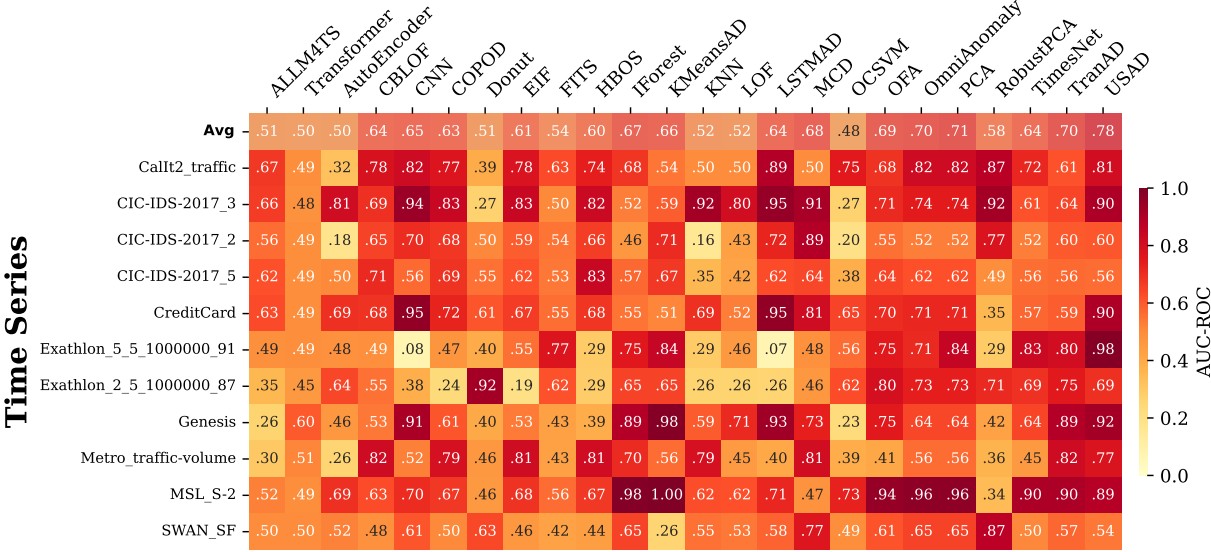

Figure E.2: **AUC-ROC (↑) of Anomaly Detectors ($x$-axis) Evaluated on MTS with Extremely Short Anomalies in mTSBench ($y$-axis).** Top row presents the average AUC-ROC over these MTS.

length is less than 20 data points. As shown in Figure E.1, semi-supervised methods MCD and OCSVM achieve the highest average AUC-ROC on long anomalies, outperforming deep learning and LLM-based approaches, despite their simplicity. For short anomalies (Figure E.2), USAD performs best, likely due to its architecture's sensitivity to localized, transient deviations. These findings highlight the robustness of classic semi-supervised methods across anomaly duration extremes. Notably, LLM-based detectors do not exhibit clear advantages over more traditional techniques in either regime.

## F    Implementation Details

**Anomaly Detection Methods.** In real-world deployments, users often rely on off-the-shelf anomaly detectors without the time, supervision, or expertise required for data-specific tuning. Effective model selection is therefore critical: given a pool of detectors, the goal is to automatically choose the one likely to perform best for a new time series. To reflect this practical constraint, we use default settings for all anomaly detectors, details of which can be found in the mTSBench repository. Model selectors operate solely by ranking detector performance, ensuring that their contribution is isolated from detector optimization. This setup allows us to evaluate selectors under realistic, low-supervision conditions where anomaly characteristics may vary widely across domains. In practice, anomaly detection is often used in real-world unsupervised settings, where true labels are not available. In such cases, threshold tuning is impractical. To reflect practical settings, we avoid method-specific threshold tuning and adopt a fixed quantile-based strategy: the top 7.5% of anomaly scores are labeled as anomalies, regardless of the detector or dataset. While this may not yield the best absolute performance per method, it ensures consistent comparisons across detectors and datasets and reflects a realistic and fair evaluation scenario, especially for benchmarking unsupervised methods.

**Model Selection Methods.** MetaOD employs PCA to compute meta-features, with the number of components originally set to 3. However, for datasets with fewer than three dimensions (excluding label and timestamp), the number of components is adjusted accordingly. This method produces a ranked list of detectors. Similarly, FMMS generates a ranking of anomaly detection methods. FMMS leverages data and meta-features from matrix factorization methods (Fusi et al., 2018); however, the code for computing these meta-features is not publicly available, nor there exist sufficient explanations of their computation process. To ensure consistency, we utilize the same meta-features as MetaOD, with no other modifications to the FMMS method. Meta-features for MetaOD and FMMS are derived by extracting statistical properties of the datasets and characteristics from various outlier detection models, including structural features and outlier scores. They capture critical dataset characteristics to identify similar datasets, thereby enhancing model selection for anomaly and outlier detection tasks. In contrast, Orthus evaluates six detectors with various configurations and employs 22 univariate time series meta-features, which are summarized into five statistical values $\{\min, Q_1, \text{mean}, Q_3, \max\}$ for each feature, resulting in a total of 110 meta-features. To accommodate smaller anomaly detection algorithm sets, we adjust the number of neighbors and components to lower values. Moreover, all model selection methods in mTSBench require a performance matrix. Rather than constructing this matrix at the individual time series level, which assumes access to labeled data for every time series, we adopt a more practical approach by building the matrix at the dataset level. We argue that in realistic settings, obtaining labeled instances for every time series solely to build a performance matrix is impractical. Moreover, if such labels were available, more direct strategies for choosing detectors could be employed. Accordingly, we construct a performance matrix of shape $19 \times 24$, representing 19 datasets and 24 detectors, for training the model selectors. Detector rankings used to evaluate model selection performance are derived from their VUS-PR scores on the test set.

**Computational Resources.** All experiments were conducted on a high-performance computing cluster using a single Intel CPU node with 36 cores at 2.3 GHz. Figure F.1 (top) shows the average runtime of each anomaly detector over 344 multivariate time series in mTSBench (for semi-supervised methods, the runtime includes both training and inference times). Among the 24 methods, IForest is the most efficient, followed by ALLM4TS and MCD, while AnomalyTransformer exhibits the highest computational cost, and OFA is the second most expensive. We further include a heatmap reporting the mean runtime of all detectors across the 19 datasets (Figure F.1 bottom). The results show no consistent correlation between dataset group and runtime efficiency. Instead, detector behavior is highly variable: a method that is efficient on one dataset may incur substantially higher cost on another. For example, PCA records its longest mean runtime on CIC-IDS-2017, where several other methods, *e.g.*, IForest, remain comparatively efficient. Conversely, IForest's largest runtimes are concentrated on SWAN-SF, OCCUPANCY, and MITDB.

## G    Limitations

While mTSBench represents the most comprehensive benchmark for MTS-AD and model selection to date, several limitations remain. The current benchmark focuses on selecting a single best-performing detector

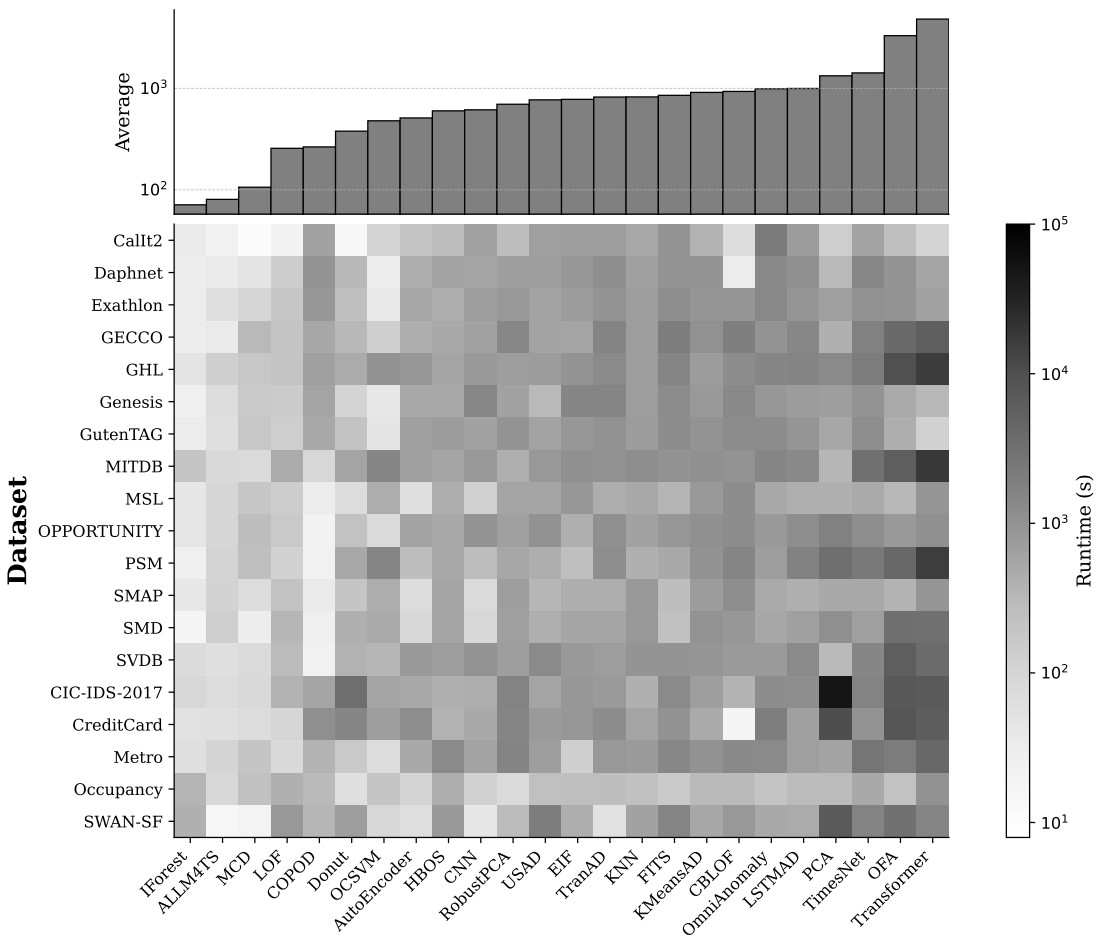

Figure F.1: **Average Runtime (↑) of Anomaly Detectors (*x*-axis) over mTSBench Datasets (*y*-axis)**. Top bars indicate per-detector average runtime over all 344 MTS in mTSBench.

per time series. Future work may explore ensemble-based selectors or context-aware mixtures of experts. In addition, although the use of default hyperparameter values mirrors practical, low-supervision deployments, it may underrepresent the potential of methods that benefit from tuning. Nevertheless, mTSBench provides an interface to modify detector configurations, enabling future work to explore the impact of tuning under more controlled settings. We also note that the datasets used in the experiments, like many publicly available time-series datasets used in anomaly detection studies, may be affected by issues documented in prior work (Wu & Keogh, 2023), such as mislabeled ground truth, unrealistic anomaly density, trivial anomaly patterns, and run-to-failure bias. For example, some ECG datasets (*e.g.*, MITDB and SVDB) contain multiple occurrences of the same arrhythmia class; however, individual events in typical ECG data differ in morphology, amplitude, duration, and multivariate expression across leads. Other dataset problems include potential unlabeled anomalous segments (*e.g.*, in MSL and Genesis), high-magnitude anomalies in certain channels (*e.g.*, in PSM, SMD, and SMAP), and simplified patterns in some synthetic datasets. These properties reflect common challenges in anomaly annotation and are not unique to mTSBench. Users may select or exclude datasets based on their requirements and tolerance for labeling noise. We point out these potential issues to support the responsible use of mTSBench and to emphasize the need for future work on more diverse, reliably annotated, and representative datasets for MTS-AD. Lastly, while mTSBench includes two LLM-based detectors, which, to the best of our knowledge, are the only publicly available such methods for MTS-AD, the space of foundation model-based approaches is rapidly evolving. Extending the benchmark to include fine-tuned LLMs or multimodal foundation models remains a promising direction for future research.

