# OpenReview forum: "mTSBench: Benchmarking Multivariate Time Series Anomaly Detection and Model Selection at Scale"
_TMLR — Accepted by TMLR_

### Review · Reviewer_Ma2w · 2025-10-19

**Summary Of Contributions:**

The authors assemble some datasets and do a "bake off"

**Additional Comments:**

I appreciate that the paper required lots of effort.

However, at the end of the day, it has assembled a bunch of datasets of dubious provenance, that are known to contain lots of false positive and false negatives, suffer from triviality, and some of the datasets have a huge problem with overcounting success (Not K anomalies, but one anomies repeated K times [a])  [b]'

The authors need to have a lot more introspection about the data, and edit out all the dubious ones (or at least warn the user, something like “this dataset may have up to 50% false negatives in the ground truth”.


1)	For almost all of SVDB and MITDB, there is a huge overcounting problem. The data is labeled as having upto 720 anomalies. But it does not, it has a single anomaly, 720 times. For almost all these datasets, in each given dataset the anomaly is a arrhythmia, but it is the SAME arrhythmia again and again. If you can detect one, you can detect them all! There is a short video that explains this [a].

2)	For SVDB and MITDB, where do the labels come from? The answer is that over 25 years ago, someone ran a simple beat labeling tool to label the anomalous beats.  But think about that, that means there is a simple, fast bit of software out there that can get 100% accuracy on these datasets! Does that say that these are too simple to be interesting?

3)	For almost all of SVDB and MITDB, they yield to Wu’s one-liner argument [b]. Does that say that these are too simple to be interesting?

4)	Metro. This  is really a classification dataset. The labels are “US National holidays plus regional holiday, Minnesota State Fair” which are not really anomalous. Yet, an entire weekend with no traffic (presumably construction or a broken senseo) is not anomalous?!? There are no annotations for major car crashes, whiteout blizzards etc. This dataset is plagued with dozens of false positives and false negatives.

5)	GHL. This datasets is too trivial to be interesting. Plot LevcorrTempfaultseed199vars23(:,6) with the ground truth labels, now plot plot(zscore((LevcorrTempfaultseed199vars23(:,6)))>1.1)  , it produced the exact same labels.

6)	GutenTAG. These datasets is too trivial to be interesting. Take a look at  GutenTAG ecg-diff-count-2  , now plot( movstd(test(:,2),25)<1.5);,   It is an embarrassingly simple pair of anomalies.  This entire dataset is embarrassing, it should never have been published.

7)	MSL. This datasets is plagued by false negatives. Let's look at G-1. The only anomaly labeled in 4770 to 4890. However, 4270 to 4285 and 6880 to 6894 are anomalies too, they were missed by the human labeler.

8)	SDM is a joke, plot it with the labels a just look at it.

9)	Fig B.1. Please us x-axis labels in such plots.

10)	In this paper you compute “average AUC-ROC”, is that meaningful when the datasets have different sizes, default rates, difficulties etc?  No, it is clearly meaningless. I know other papers do it, but it is still meaningless.

[a] https://www.youtube.com/watch?v=fR4vqwmALZM

[b] Current Time Series Anomaly Detection Benchmarks are Flawed and are Creating the Illusion of Progress. IEEE Trans. Knowl. Data Eng. 35(3): 2421-2429 (2023)

**Audience:**

Yes

**Audience Explanation:**

See main review.

**Claims And Evidence:**

No

**Claims Explanation:**

The datasets used are deeply flawed, which "poisons" any conclusions.

**Requested Changes:**

I don't think this paper can be saved.

Producing a single good multiD TSAD datasets would be worthy of a paper. But it seems like no one is able or willing to do it.

---

> ### Author Response · Authors · 2025-11-29
>
> We appreciate the reviewer’s comment regarding dataset quality and clarify that mTSBench’s goal is to provide a transparent and extensible framework for evaluating anomaly detection, where more datasets, anomaly detection methods, and model selection techniques can be easily integrated. While dataset quality inevitably varies, our framework isolates this variability and allows researchers to focus on relative detector performance under standardized conditions. Researchers can choose which datasets or time series to include in their studies based on their specific use cases and domains of interest. Label refinement and domain-specific relabeling are valuable future community efforts that require deep domain expertise related to the original data sources, and are *beyond* the scope of this benchmark’s core contribution.
>
> Regarding the specific datasets mentioned:
>
> * (1) Although some datasets (e.g., SVDB, MITDB) may appear simple, our results (Fig. 1) show that state-of-the-art detectors still struggle on them, indicating that these datasets remain non-trivial for current anomaly detection methods.
> * (2) All datasets in mTSBench are *multivariate*. Some examples (e.g., Metro, GHL, MSL, SMD) seem to have been interpreted as *univariate*. For instance, GHL includes 16 dimensions, all provided to the detector without prior knowledge of which are relevant. Although a human *might* identify the anomaly by inspecting a single dimension, the detector receives all dimensions jointly and must infer which ones are anomalous and how they interact, which remains a non-trivial problem in multivariate settings.
> * (3) Regarding GutenTAG, we adapted the original implementation to generate 30 diverse and more challenging time series, whose anomalies cannot be detected by simple thresholding. Implementation details are provided in Appendix B.
>
> The absence of perfect datasets should not hinder progress in anomaly detection research. Nonetheless, we have now documented in our paper the known limitations of existing datasets to support transparent and responsible use of mTSBench.

---

> > ### Comment · Reviewer_Ma2w · 2025-11-29
> > **I appreciate the response to my points, but still have concerns about data quality**
> >
> > I appreciate the response to my points and in particular I appreciate  "...we have now documented in our paper the known limitations of existing datasets to support transparent and responsible use of mTSBench." and I am raising my opinion slightly.
> >
> >
> >
> > However, I still think that most of these datasets are meaningless
> >
> > 1)	Consider SVDB, MITDB
> > A)	The labels created by a simple 25 year old beat detector, implying this is easy.
> > B)	The 25 year old beat detector, was looking for particular heart arrhythmias. However there are other anomalies of noise, movement artifacts and other arrhythmias that were not flagged, that means there are false negatives in the ground truth.
> > C)	You evaded my critical point “For almost all of SVDB and MITDB, there is a huge overcounting problem. The data is labeled as having upto 720 anomalies. But it does not, it has a single anomaly, 720 times. For almost all these datasets, in each given dataset the anomaly is a arrhythmia, but it is the SAME arrhythmia again and again. If you can detect one, you can detect them all!
> >
> >
> > 2)	Consider SMD [1], PSM  [2] and SMAP, many of the anomalies are when the mean or standard deviation changes by one, two, three even four orders of magnitude. You can find these with simple statistical process control algorithms. Does it make sense  to compare complex algorithms on them?
> >
> >
> > 3)	Consider Metro, it is plagued with bad labels, There is an entire weekend with zero traffic, but that is not marked as an anomaly.  Look very carefully at this dataset with the labels, the labels appear to be close to random.
> >
> >
> >
> > 4)	Consider Genesis. There are regions that are completely constant, but within that constant the labels toggle from anomaly to not-anomaly multiple times.  There is no provenance (that I am aware of)  to contradict  the obvious conclusion that the labels have errors.
> >
> >
> > You (and to be fair, most of the community) seem unwilling to deal with the fact that most datasets are trivial and/or so badly labeled as to be meaningless. I have some sympathy for this, getting high quality datasets IS hard work. But personally, I have close to zero faith in results obtained on such datasets, and for that reason I cannot champion your paper.
> >
> >
> > You ignored my point “In this paper you compute “average AUC-ROC”, is that meaningful when the datasets have different sizes, default rates, difficulties etc.? ” That is fine, but it is still obviously true.
> >
> > [1]
> > https://github.com/boniolp/ADecimo/blob/main/data/benchmark_ts/SMD/machine-1-1.test.csv%4010.out.zip
> >
> > [2]
> >  There is an anomaly marked at 74450, but look at column 23. There is a spike there that is an order of magnitude larger than the surrounding data.
> >
> > There is an anomaly marked at 32510, but look at column 23. There is a spike there that is two orders of magnitude larger than the surrounding data.
> >
> > There is a cluster of three anomalies marked at 10440, 10460 and 10520, but look at column 22. There is a spike there that is four orders of magnitude larger than the surrounding data.
> >
> > Look at the anomaly region from 7063 to 7398, now look at column 18 in the same region, how could ANY method fail to find these?
> >
> > [3] https://github.com/boniolp/ADecimo/blob/main/data/benchmark_ts/Genesis/genesis-anomalies.test.csv%406.out.zip

---

> > > ### Author Response · Authors · 2025-12-01
> > >
> > > We appreciate the reviewer’s continued comments, and we fully agree that constructing high-quality multivariate anomaly detection datasets is extremely challenging. This point has been repeatedly a focus in our own discussions with prominent researchers in this field. Labeling anomalies in real multivariate systems is inherently noisy, costly, and often ambiguous, and in many cases, identifying multivariate interactions requires deep domain expertise in respective application domains. These constraints make it difficult for the community to maintain perfectly curated anomaly detection datasets.
> > >
> > > In response to the reviewer’s concerns, we have now made the dataset limitations more explicit in the paper to document known issues (label noise, overcounting, trivial anomalies, and missing events), and provide clear warnings so that users understand the provenance and constraints of each dataset. Our intention is not to claim these datasets are flawless, but to acknowledge their imperfections and present mTSBench as a transparent, extensible framework where each dataset’s limitations are openly described, and where users can include, exclude, or replace datasets depending on their specific goals.
> > >
> > > We have also taken a closer look at several of the reviewer’s points with additional analysis. For example, in MITDB/SVDB, the reviewer notes the presence of repeated arrhythmia types. We examined the per-event variability and found that individual occurrences differ in morphology, duration, amplitude, and multivariate expression across ECG leads. Consequently, we observe that anomaly detectors do not detect all occurrences uniformly. An illustrative subset of this analysis is provided in the table below. The results show large variability across methods: some detectors, such as TimesNet and LSTMAD, successfully recover most ground-truth events but still produce a substantial number of false-positive anomaly sequences, while others, including OmniAnomaly, ALLM4TS, and OFA, miss the majority of events altogether.
> > >
> > > | Detector           |   Hit Rate |   # False Positives |
> > > |:-------------------|---------------------------:|----------------------:|
> > > | OmniAnomaly        |                      0.138 |                   129 |
> > > | AnomalyTransformer |                      0.724 |                  1995 |
> > > | LSTMAD             |                      0.793 |                  1983 |
> > > | TimesNet           |                      1     |                  1799 |
> > > | ALLM4TS            |                      0.103 |                   115 |
> > > | OFA                |                      0.069 |                   148 |
> > >
> > > We have also tried to carefully examine the PSM dataset. We agree that some anomalies correspond to large changes in mean or variance, including the example highlighted by the reviewer (indices 7063–7398 in feature 18 of PSM, which appears to be part of the validation split). However, our visual inspection of features 18 and 22 of PSM shows a mixture of anomaly types: some are large shifts, while others are shorter and subtler. To our understanding, in PSM, anomalies are annotated as event intervals based on system-level operational logs, and as a result, a spike in one channel may not necessarily correspond to an anomaly; many spikes seem to reflect normal multivariate behavior when viewed across all dimensions. Conversely, some true anomalies manifest only when considering cross-feature interactions.
> > >
> > >
> > > Regarding evaluation metrics in mTSBench, AUC-ROC is one of 13 evaluation metrics we report, indeed following standard practice in time-series anomaly detection benchmarks. Our use of AUC-ROC serves as one normalized, threshold-independent ranking measure used alongside a broad suite of point-wise and range-based metrics. Our analyses and conclusions do not rely on averaged AUC-ROC alone but are supported by multiple complementary metrics.
> > >
> > > We sincerely thank the reviewer for highlighting these issues and believe the revisions have substantially improved the transparency and usability of mTSBench.

---

> > > > ### Comment · Reviewer_Ma2w · 2025-12-01
> > > >
> > > > Thanks for you nice discussion of PSM which I think is 100% correct.
> > > >
> > > > I think you may have slightly missed my point on MITDB/SVDB, so I have expanded it below.
> > > >
> > > > It is lot to read, sorry!
> > > >
> > > >
> > > > Consider MITDB dataset, I grabbed the last one 234 at random.
> > > > It happens to only have three Premature ventricular contractions, lets plot them
> > > > >> plot( ekg1(  369391 : 369391 + 300,3));
> > > > >>hold on;
> > > > >> plot( ekg1(  464997 : 464997 + 300,3));
> > > > >> plot( ekg1(  615531 : 615531 + 300,3));
> > > > They are basically identical. I would not mind a TSAD algorithm taking credit for finding one of these. But if you take credit for “we got 3 out of 3!”, then you are overcounting.
> > > >
> > > > ---
> > > >
> > > > Later, there is a long run of Nodal premature beats. Lets compare them to the normal beats that happen a few seconds later.
> > > > >> figure;   hold on;, plot( ekg1(  312304 : 312304+ 300,2),'r');, plot( ekg1(  312457 : 312457+ 300,2),'r');,  plot( ekg1(  312605 : 312605+ 300,2),'r'); % Nodal (junctional) premature beat
> > > > >>  plot( ekg1(  317099 : 317099+ 300,2),'g');, plot( ekg1(  317342 : 317342+ 300,2),'g');,  plot( ekg1(  317582 : 317582+ 300,2),'g'); % normal beats a few seconds later
> > > >
> > > >
> > > > Two things jump out.
> > > > A)	This looks easy
> > > > B)	The anomalies (Nodal premature beats) all look basically identical.
> > > >
> > > > --
> > > >
> > > > Now I grabbed the second-last one, 233..
> > > > There are 2234 normal beats.
> > > > There are 831 V beats.
> > > > Is a 37% anomaly density meaningful?
> > > >
> > > > Lets plot some anomalies and some normals.
> > > >
> > > > >> figure;   hold on;, plot( ( ekg2( 1107: 1107+300 ,2)),'r');, plot( ( ekg2( 42: 42+300 ,2)),'r');,  %anomaly
> > > > >> plot( ( ekg2( 2446: 2446+300 ,2)),'b');, plot( ( ekg2( 2671: 2671+300 ,2)),'b');, %normal
> > > > Two things jump out.
> > > > A)	This looks very easy
> > > > B)	The anomalies all look basically identical.
> > > > I will concede that there are two anomaly types here, V and A. And you might even argue that V has some slight variance due to wandering baseline and to slightly changing heartrate. But if you are taken credit for  hundreds of detections, that is clearly overcounting.
> > > > --
> > > > You are using the beat labels as ground truth. But there are other anomalies that in the data, but not annotated, for example this noisy normal beat
> > > > >> plot( ( ekg2( 285546: 285546+300 ,2)),'b');,
> > > > Likewise, while invariance to wandering baseline is generally good, there are some severe movement artifacts in the data. At least in some cases, these should be flagged as anomalies. At a minimum, there should be some domain introspection and disclosure about this.
> > > > ---
> > > > To be clear, I am not saying that ECGs could never be used for TSAD. In fact, there is a great example that Matlab has [mat]. There is exactly one anomaly (ventricular tachycardia) so no overcounting. Matlab did a great job here.
> > > >
> > > > ---
> > > >
> > > > To summarize: I worry about using MITDB/SVDB because
> > > >
> > > > 1)	Huge amount of overcounting.
> > > > 2)	Triviality, we are reproducing indirectly, algorithms that we had 25+ years ago.
> > > > 3)	Greatly unrealistic anomaly density.
> > > > 4)	Errors in the ground truth, because the labels do not including noise, movement artifacts, dropouts and other “non-beat” anomalies.
> > > > 5)	An intrinsic mismatch of purpose, this is a trivial classification problem, not an TSAD problem.
> > > >
> > > >
> > > > [mat] Case 2: Detect Anomalous Points in Continuous Long Time Series
> > > > https://www.mathworks.com/help/deeplearning/ug/detect-anomalies-in-signals-using-deep-learning.html

---

> > > > > ### Author Response · Authors · 2025-12-02
> > > > >
> > > > > We thank the reviewer for the detailed comments on the MITDB and SVDB datasets. We have incorporated descriptions of these characteristics into the revised manuscript.
> > > > >
> > > > > Thank you for your time and effort invested in our work.

---

> > > > > > ### Comment · Reviewer_Ma2w · 2025-12-02
> > > > > > **Thank you for being responsive:  Two minor points: (but no action needed)**
> > > > > >
> > > > > > Wonderful. Thank you for being responsive
> > > > > >
> > > > > > Two minor points: (no action needed)
> > > > > >
> > > > > > 1)	“Note that the time series in this dataset MAY contain trivial anomalies that are similar to one another, and the performance of detectors on MITDB should be interpreted with caution.” (my emphasis).  I am not sure why you qualify with “may”. In most cases, they “do” contain literally hundreds (per each recording). I pointed out that the exact locations of some, and five minutes of work would let you find 1,000’s of examples of anomalies that are similar (essentially IDENTICAL) to one another. You also do not mention the fact that these files have many false negatives.
> > > > > >
> > > > > > 2)	The anomalies’ locations given in the files are a function of how the beat extractor worked 26 years ago. The beat extractor is designed to point to a  local maximum or a minimum. However, these are not the locations that a cardiologist would point to as the beginning of an anomaly. In almost all cases the  cardiologist could see only some data before the annotation and know it was an anomaly.  This is critical and underappreciated, because many scoring functions treat the annotated location as “gospel” and would (unfairly) penalize a method that predicted an anomaly a little earlier.

---

> ### Author Response · Authors · 2025-12-02
>
> We appreciate your quick response and we have revised the sentence according to your kind suggestions. Thank you!

---

### Review · Reviewer_D2mp · 2025-10-21

**Summary Of Contributions:**

This paper's primary contribution is the introduction of mTSBench, a benchmark designed to tackle the critical challenge of model selection in multivariate time series anomaly detection (MTS-AD). By systematically evaluating a large collection of 344 time series, the benchmark not only underscores the significant performance variability among existing detectors but also delivers the key finding that current model selection strategies are substantially suboptimal, revealing critical gaps in their ability to handle complex temporal dynamics. To catalyze future research, mTSBench provides a unified evaluation suite that establishes a foundation for reproducible work and motivates new directions in adaptive, context-aware model selection and the integration of foundation models.

**Audience:**

Yes

**Audience Explanation:**

The findings presented are highly relevant to the TMLR audience as they address the critical, real-world problem of model selection for multivariate time series anomaly detection, which is a major deployment bottleneck. The paper provides a timely shift in focus from proposing new detectors to solving the model selection challenge itself. It offers valuable early insights into emerging LLM-based methods and, by open-sourcing a comprehensive benchmark, establishes an essential foundation for future research in automated and adaptive machine learning for time series.

**Broader Impact Concerns:**

The paper's broader impact statement is well-considered, highlighting positive societal benefits through improved robustness in safety-critical domains and acknowledging risks like missed detections or false alarms from poorly-validated selectors. The benchmark itself mitigates these risks by exposing current limitations. A valuable addition would be to explicitly address the computational and environmental costs of model selection, particularly for LLM-based methods, and to encourage future work on efficient as well as accurate selectors.

**Claims And Evidence:**

Yes

**Claims Explanation:**

1.	The introduction of mTSBench addresses a critical and underexplored gap in the field: standardized benchmarking for model selection in multivariate time series anomaly detection (MTS-AD). Its scale (344 time series, 19 datasets, 24 detectors) and focus on model selection make it a comprehensive resource of its kind.
2.	The empirical evaluation is exceptionally thorough. The use of 13 diverse evaluation metrics for anomaly detection and 3 for model selection, combined with comparisons against strong baselines (Oracle, Near-optimal, Random), provides a robust and multi-faceted assessment of performance.
3.	The key results are clearly demonstrated: no single detector is universally superior, and more importantly, current model selection methods remain significantly sub-optimal. These findings provide a powerful motivation for future research, while the performance analysis across dimensions, anomaly lengths, and domains offers nuanced insights.
4.	The paper is well-structured and clearly defines the problem, methodology, and evaluation criteria. The extensive appendices detailing datasets, detector implementations, and meta-feature extraction greatly enhance reproducibility and trust in the findings.

**Requested Changes:**

1.	The benchmark is currently restricted to three specific model selection methods (MetaOD, FMMS, Orthus). Please clarify why choose these three methods. Other promising paradigms, such as reinforcement learning-based selection or more sophisticated ensemble strategies (beyond selecting a single best model), are not explored.
2.	Please describe the relationship and distinction between the three trivial baselines and the three model selection methods presented in Table 2. Additionally, the data for "Oracle" and "Near-optimal" are highlighted with a gray background. What does this formatting signify?

---

> ### Author Response · Authors · 2025-10-26
>
> We thank the reviewer for the thoughtful and encouraging feedback, and appreciate the recognition of mTSBench’s clarity, scale, and contribution to advancing research on model selection in multivariate time series anomaly detection.
>
> For model selection baselines, we followed a systematic approach to include methods with (a) publicly available and executable code, (b) compatibility with multivariate time series detectors, and (c) no dependence on test labels or test-time detector runs. To the best of our knowledge, 13 selection methods can be applied to selecting anomaly detectors, as discussed in the related work section of the paper. Of these, five lack publicly available implementations. In addition, the following methods were excluded due to scope limitations:
>
> * **MOSPAT**: tightly integrated into a proprietary univariate library, restricted to univariate detectors and thus incompatible with multivariate detectors.
> * **TSADMS**: requires running all anomaly detectors in the selection pool on test data to make selection decisions (defeats the purpose of model selection).
> * **ELECT**: incomplete code prevented execution despite our multiple attempts.
> * **IMES**: reports its internal evaluation design as ineffective for model selection [1].
> * **RLMS**: requires running all detectors on test data for state updates (defeats the purpose of model selection).
>
> The three trivial baselines (Oracle, Near-optimal, Random) are included to establish performance bounds. Specifically, the baselines do not rely on any learning or meta-features. They either assume access to ground-truth anomaly labels (Oracle, Near-optimal) or perform selection uniformly at random (Random). In contrast, the three model selection methods aim to automatically choose the best detector for each time series without access to anomaly labels, using learned relationships or meta-information. The gray shading in Table 2 separates Oracle and Near-optimal from the rest, as their performance is achieved using ground-truth rankings of the detectors, and they are included solely as performance references rather than feasible unsupervised methods.
>
> Regarding broader impacts, we agree that computational and environmental costs are important considerations for large-scale model selection, particularly as methods increasingly rely on deep and LLM-based architectures. We have revised the broader impact statement, acknowledging that efficiency and sustainability are key directions for future work.
>
> [1] Ma, Martin Q., et al. "A large-scale study on unsupervised outlier model selection: Do internal strategies suffice?." arXiv preprint arXiv:2104.01422 (2021).

---

### Review · Reviewer_gyyr · 2025-11-15

**Summary Of Contributions:**

The authors present a new benchmark for anomaly detection of time series (i.e. detecting which points in a time series are potential anomalies). The benchmark is orignal for two reasons:
- its scale (in terms of numbers of datasets, competitors, etc.): it is slightly larger than existing benchmarks,
- they include a benchmark of model selection techniques, which are often overlooked in this context.

The authors also draw insights on anomaly detection for time series, and in particular highlight the importance of model selection and conclude that there is no absolute best method. An interesting conclusion of their benchmark is also that a simple PCA-based baseline is amongst the strongest competitors.

**Audience:**

Yes

**Audience Explanation:**

I think model selection in this context is a somewhat overlooked and important problem, thus this benchmark could be quite useful to the community. The benchmark contains nicely diverse datasets, methods, and evaluations.

The main two limitations are the use of default hyperparameters (model selection should precisely be useful to choose hyperparameters, so looking into this would seem very relevant) and the lack of a basic ensemble baseline.

**Claims And Evidence:**

Yes

**Claims Explanation:**

The main claims are that
- the benchmark is the largest existing one, which appears to be true, although the scale is not dramatically larger than TSB-AD-M
- the benchmark is the first one to systemically look at model selection. While this appears to be true for multivariate time series, I think that the authors should acknowledge more clearly that such bechmarks exist for univariate time series.

**Requested Changes:**

These changes are ordered by importance. The first two are critical, and the last one would significantly improve the insights provided by the benchmark, but is less critical.

1. In the "related works" section, the authors devote one subsection to "Model Selection for Time Series" and one to "Anomaly Detection Benchmarks". It would be important to indicate that the intersection between these two is non-empty for univariate time series, and includes at least a benchmark cited by the authors (Sylligardos et al., 2023). Discussing the relationship between the findings and design choices of the authors and this benchmark would improve the paper. I am not an expert of time series, and I am not aware of other benchmarks than Sylligardos et al. (2023) but if other benchmarks for model selection exist in the univariate case, they should be discussed as well.

2. I think one of the main findings of the paper, that resonates with other works (in particular TSB-AD) is that simple methods like PCA are very strong baselines. This should be emphasized more. In particular, it would be interesting to discuss how PCA fares agains the model selection baselines.

3. I think adding a basic ensembling baseline against model selection methods would be insightful. This is a very natural alternative to model selection, and it was done for instance in the univariate benchmark of Sylligardos et al. (2023).

---

> ### Author Response · Authors · 2025-11-29
>
> We thank the reviewer for their constructive and thoughtful feedback. We appreciate the recognition of mTSBench’s scale, diversity, and contribution to advancing research on model selection for multivariate time series anomaly detection.
> Regarding related work, Sylligardos et al. (2023) evaluate whether time series classifiers can serve as model selectors for anomaly detectors. While this is a highly valuable contribution, it differs from benchmarking dedicated model selection methods. Our work aims to fill this gap by evaluating established model selectors (e.g., feature-based, similarity-based, or meta-learning approaches) in multivariate settings. We have revised the related work section to clarify this point.
>
> Regarding simple baselines, we thank the reviewer for highlighting this point. We agree that one of the **most interesting takeaways is that simple methods like PCA are quite competitive**. In the revised manuscript, we now explicitly compare PCA against model selection methods in Table 2 (also shared below) and discuss this in Section 4.2. We observe that PCA outperforms all three selectors (MetaOD, FMMS, Orthus) on 9 out of 13 anomaly-detection metrics, but performance is still clearly below the Near-optimal and Oracle references. This reinforces our conclusion that existing model-selection techniques remain substantially suboptimal, and that model selection remains a fundamental yet unexplored open problem in MTS-AD.
>
> Regarding the reviewer’s insightful suggestion for an **ensemble baseline**, we now report the performance of basic ensemble strategies on mTSBench in Table 2 (also shared below). The ensemble baseline does worse than FMMS and MetaOD on all metrics, and is only similar on AUC-ROC and VUS-ROC.
> | Metric               | Oracle | Near-optimal | Random | PCA | MetaOD | FMMS | Orthus | Ensemble |
> |----------------------|------------------|---------------|--------|-----|--------|--------|---------|-----------|
> | F1                   | 0.546 ± 0.263    | 0.514 ± 0.256 | 0.299 ± 0.252 | 0.252 ± 0.195 | 0.135 ± 0.135 | 0.222 ± 0.185 | 0.222 ± 0.200 | 0.189 ± 0.187 |
> | Precision            | 0.414 ± 0.328    | 0.393 ± 0.319 | 0.194 ± 0.236 | 0.293 ± 0.305 | 0.189 ± 0.236 | 0.280 ± 0.299 | 0.297 ± 0.311 | 0.237 ± 0.262 |
> | Recall               | 0.483 ± 0.298    | 0.458 ± 0.287 | 0.275 ± 0.313 | 0.319 ± 0.310 | 0.205 ± 0.210 | 0.307 ± 0.276 | 0.283 ± 0.271 | 0.300 ± 0.301 |
> | Affiliation-F1       | 0.879 ± 0.107    | 0.868 ± 0.101 | 0.785 ± 0.124 | 0.818 ± 0.108 | 0.777 ± 0.100 | 0.834 ± 0.112 | 0.814 ± 0.111 | 0.792 ± 0.107 |
> | Event-based-F1       | 0.699 ± 0.287    | 0.667 ± 0.283 | 0.427 ± 0.338 | 0.506 ± 0.329 | 0.377 ± 0.305 | 0.548 ± 0.333 | 0.500 ± 0.340 | 0.433 ± 0.330 |
> | Max-F1               | 0.546 ± 0.263    | 0.514 ± 0.256 | 0.299 ± 0.252 | 0.408 ± 0.268 | 0.292 ± 0.246 | 0.381 ± 0.263 | 0.386 ± 0.270 | 0.379 ± 0.270 |
> | PA-F1                | 0.825 ± 0.235    | 0.803 ± 0.226 | 0.705 ± 0.316 | 0.627 ± 0.320 | 0.658 ± 0.324 | 0.738 ± 0.285 | 0.647 ± 0.332 | 0.584 ± 0.332 |
> | R-based-F1           | 0.450 ± 0.237    | 0.430 ± 0.224 | 0.252 ± 0.190 | 0.410 ± 0.237 | 0.242 ± 0.194 | 0.366 ± 0.224 | 0.370 ± 0.236 | 0.328 ± 0.246 |
> | AUC-PTRT             | 0.417 ± 0.276    | 0.400 ± 0.260 | 0.234 ± 0.228 | 0.322 ± 0.278 | 0.217 ± 0.218 | 0.318 ± 0.273 | 0.309 ± 0.272 | 0.278 ± 0.260 |
> | AUC-PR               | 0.492 ± 0.296    | 0.455 ± 0.284 | 0.212 ± 0.222 | 0.327 ± 0.280 | 0.210 ± 0.231 | 0.304 ± 0.266 | 0.315 ± 0.279 | 0.287 ± 0.271 |
> | AUC-ROC              | 0.811 ± 0.150    | 0.796 ± 0.148 | 0.613 ± 0.178 | 0.697 ± 0.219 | 0.605 ± 0.189 | 0.659 ± 0.207 | 0.672 ± 0.219 | 0.683 ± 0.214 |
> | VUS-PR               | 0.524 ± 0.296    | 0.485 ± 0.290 | 0.236 ± 0.236 | 0.348 ± 0.290 | 0.229 ± 0.231 | 0.326 ± 0.277 | 0.333 ± 0.288 | 0.312 ± 0.279 |
> | VUS-ROC              | 0.847 ± 0.134    | 0.832 ± 0.134 | 0.640 ± 0.188 | 0.721 ± 0.211 | 0.655 ± 0.179 | 0.698 ± 0.204 | 0.707 ± 0.212 | 0.714 ± 0.203 |

---

> ### Author Response · Authors · 2025-11-29
>
> Finally, in terms of **hyperparameter selection**, we ran additional experiments in which model selectors chose among 42 PCA hyperparameter settings. Because all selectors operate on the same candidate pool and under the same evaluation setting, their relative behavior remained consistent with our main detector-selection results: Orthus performed best, followed by FMMS. Although increasing the pool enlarges the search space, it does not change the comparative setting across selectors, and empirically, we observed that selector rankings remained stable.
>
> | Metric            | MetaOD              | FMMS               | Orthus                     |
> |-------------------|---------------------|--------------------|----------------------------|
> | **F1**            | 0.130 ± 0.172       | 0.154 ± 0.173      | **0.160 ± 0.169**          |
> | **Precision**     | 0.166 ± 0.235       | 0.212 ± 0.252      | **0.217 ± 0.252**          |
> | **Recall**        | 0.207 ± 0.286       | 0.225 ± 0.276      | **0.239 ± 0.285**          |
> | **Affiliation-F1**| 0.638 ± 0.351       | 0.781 ± 0.171      | **0.789 ± 0.167**          |
> | **Event-based-F1**| 0.390 ± 0.360       | 0.470 ± 0.332      | **0.477 ± 0.331**          |
> | **Max-F1**        | 0.282 ± 0.278       | 0.334 ± 0.262      | **0.335 ± 0.259**          |
> | **PA-F1**         | 0.549 ± 0.398       | 0.673 ± 0.317      | **0.677 ± 0.314**          |
> | **R-based-F1**    | 0.263 ± 0.246       | 0.314 ± 0.216      | **0.324 ± 0.227**          |
> | **AUC-PTRT**      | 0.215 ± 0.251       | 0.261 ± 0.241      | **0.264 ± 0.244**          |
> | **AUC-PR**        | 0.213 ± 0.258       | **0.257 ± 0.256**  | 0.256 ± 0.253              |
> | **AUC-ROC**       | 0.647 ± 0.187       | **0.665 ± 0.187**  | 0.654 ± 0.202              |
> | **VUS-PR**        | 0.230 ± 0.270       | 0.273 ± 0.264      | **0.280 ± 0.266**          |
> | **VUS-ROC**       | 0.675 ± 0.184       | **0.699 ± 0.171**  | 0.693 ± 0.192              |
>
> We very much appreciate the reviewer’s careful assessment and constructive feedback. We hope our additional experiments clarify the reviewer's questions, and we sincerely thank all reviewers for considering our responses when forming their final evaluation.

---

### Review · Reviewer_z1GP · 2026-01-11

**Summary Of Contributions:**

This paper introduces mTSBench, the largest benchmark to date for multivariate time series anomaly detection (MTS-AD) with explicit support for unsupervised model selection. The benchmark aggregates 344 labeled multivariate time series from 19 datasets across 12 domains and evaluates 24 anomaly detectors, spanning statistical, classical ML, deep learning, and two publicly available LLM-based methods.

A key contribution is the first systematic benchmarking of unsupervised model selection methods for MTS-AD, comparing MetaOD, FMMS, and Orthus under a unified protocol against strong baselines (Oracle, Near-optimal, Random). The paper further introduces a comprehensive evaluation suite with 13 anomaly detection metrics and ranking-based selection metrics.

Empirically, the results show substantial performance variability across datasets and confirm that no detector or selector consistently dominates, highlighting a clear and well-quantified gap between current methods and optimal selection. Overall, the work provides a valuable, reproducible benchmark and identifies an important open problem in multivariate time series analysis.

**Additional Comments:**

This is a careful and well-executed benchmark paper that fills a clear gap in MTS-AD research. The experimental design is rigorous, the analysis is thorough, and the conclusions are appropriately scoped. With minor clarifications, the work meets TMLR’s standards for soundness, relevance, and reproducibility.

**Audience:**

Yes

**Audience Explanation:**

The paper is directly relevant to researchers in time series analysis, anomaly detection, model selection, and benchmarking. Its scale, methodological rigor, and inclusion of LLM-based detectors make it particularly timely and of broad interest to the TMLR community.

**Broader Impact Concerns:**

The paper raises no significant ethical concerns. It focuses on benchmarking and evaluation rather than deployment. The Broader Impact Statement appropriately notes risks of false alarms or missed anomalies in high-stakes domains and emphasizes the need for robust validation, transparency, and domain awareness. No additional impact discussion is required.

**Claims And Evidence:**

Yes

**Claims Explanation:**

The claims are supported by extensive experiments across a diverse benchmark, with clearly defined protocols, strong baselines, and multiple complementary metrics. Importantly, conclusions are framed relative to Oracle and Near-optimal references, avoiding overclaiming and making the evidence convincing and transparent.

**Requested Changes:**

**Critical**
1. **Clarify scope of conclusions about model selection**.

    The paper should more explicitly state that the reported limitations apply to currently evaluated unsupervised selectors, not to supervised, domain-informed, or future adaptive approaches.

**Non-Critical**
1. **Strengthen failure-mode discussion**.

    A brief synthesis explaining why selectors struggle (e.g., limited temporal meta-features, sensitivity to anomaly density or length) would improve interpretability.
2. **Add short practical guidance**.

    A concise paragraph summarizing implications for practitioners (e.g., when simple baselines outperform selectors, or when ranking is preferable to top-1 choice) would strengthen applied relevance.

---

### Comment · Reviewer_Ma2w · 2025-11-15
**I do think my fellow reviewers have been far too kind to this paper.**

I do think my fellow reviewers have been far too kind to this paper.

The authors basically grabbed a bunch of public datasets without any attempt at meaningful curation, or any deep introspection.

It is well known (at least a few dozen papers mention this) that many of these datasets are deeply flawed, suffering from the problems of:
1)	Triviality
2)	Mislabeling (in some cases, by huge amounts)
3)	Redundancy (we have ONE anomaly that happens again and again)
4)	Run to Failure bias
5)	Unrealistic anomaly density

These flaws mean that any measures of success computed on these datasets are worthless, the uncertainty is greater than the differences claimed.
We should reward papers that create a well-curated data set with good provenance. But this is just a grab bag.

---

### Author Response · Authors · 2025-11-29
**We thank all reviewers for their constructive and thoughtful feedback.**

We thank all reviewers for their constructive and thoughtful feedback, as well as for recognizing **mTSBench’s** clarity, scale, and contribution to advancing research on model selection for multivariate time series anomaly detection. We appreciate the positive comments highlighting the benchmark’s systematic design, extensibility, and the importance of the model selection perspective introduced in this work.

We have addressed the main concerns raised by the reviewers through both **new experiments and clarifications**. Specifically:

* We conducted additional experiments evaluating model selectors on **PCA hyperparameter selection**, confirming the same ranking trends observed in detector selection (Orthus > FMMS > MetaOD).
* We also evaluated an **ensemble baseline**, which consistently performs worse than FMMS and MetaOD across all metrics, except for AUC-ROC and VUS-ROC, where its performance is comparable.
* To address questions about dataset quality and provenance, we clarified that mTSBench’s goal is not to redefine or relabel existing datasets, but to provide a transparent and reproducible framework where researchers can evaluate detectors and selectors under standardized conditions and easily integrate new datasets or methods. Known limitations of existing datasets are now more explicitly documented in our paper.
* We further **revised the related work section** to better differentiate our contributions from prior studies such as Sylligardos et al. (2023), which focus on evaluating classifiers as selectors rather than benchmarking established selector methods.
* We also **clarified the inclusion criteria** for model selectors, the rationale for excluding certain selectors, and the role of Oracle and Near-optimal references.
* Finally, we **expanded our broader impacts statement** to discuss the computational and environmental implications of large-scale model selection.

We very much appreciate the time and effort all reviewers have invested in our work and sincerely thank you for considering our responses.

---

### Decision · Action_Editor_nDyA · 2026-01-11

**Recommendation:** Accept with minor revision

**Additional Comments:**

The authors can work on the following aspects to further improve the quality of this paper.

1. Address the latest comments from Reviewer z1GP.

2. Include an additional section to discuss the impact of the dataset quality.

3. Select one to two datasets that are considered deeply flawed and one to two relatively clean or well-curated datasets, and present comparative analyses. This could include (a) visualizations illustrating data characteristics and anomalies, (b) detailed anomaly detection performance comparisons, and (c) an in-depth discussion of how dataset issues affect results and how these datasets could be improved. Such an analysis would help readers better understand the role of data quality in the reported findings.

**Audience:**

Yes

**Audience Explanation:**

The paper tackles the underexplored yet practically important problem of model selection for multivariate time series anomaly detection. It is likely to be useful to both researchers and practitioners in time series analysis, anomaly detection, model selection, and benchmarking.

**Claims And Evidence:**

Yes

**Claims Explanation:**

The claims are supported by extensive and carefully designed experiments, with clear protocols, strong baselines, and comprehensive evaluation using multiple complementary metrics.